# Situat3DChange: Situated 3D Change Understanding Dataset for Multimodal Large Language Model

**Ruiping Liu**[1]     **Junwei Zheng**[1]     **Yufan Chen**[1]     **Zirui Wang**[1]     **Kunyu Peng**[1]
**Kailun Yang**[2]     **Jiaming Zhang**[1,2,3,*]     **Marc Pollefeys**[3]     **Rainer Stiefelhagen**[1]
[1] Karlsruhe Institute of Technology (KIT)     [2] Hunan University     [3] ETH Zurich

## Abstract

Physical environments and circumstances are fundamentally dynamic, yet current 3D datasets and evaluation benchmarks tend to concentrate on either dynamic scenarios or dynamic situations in isolation, resulting in incomplete comprehension. To overcome these constraints, we introduce Situat3DChange, an extensive dataset supporting three situation-aware change understanding tasks following the perception-action model: $121K$ question-answer pairs, $36K$ change descriptions for perception tasks, and $17K$ rearrangement instructions for the action task. To construct this large-scale dataset, Situat3DChange leverages $11K$ human observations of environmental changes to establish shared mental models and shared situational awareness for human-AI collaboration. These observations, enriched with egocentric and allocentric perspectives as well as categorical and coordinate spatial relations, are integrated using an LLM to support understanding of situated changes. To address the challenge of comparing pairs of point clouds from the same scene with minor changes, we propose SCReasoner, an efficient 3D MLLM approach that enables effective point cloud comparison with minimal parameter overhead and no additional tokens required for the language decoder. Comprehensive evaluation on Situat3DChange tasks highlights both the progress and limitations of MLLMs in dynamic scene and situation understanding. Additional experiments on data scaling and cross-domain transfer demonstrate the task-agnostic effectiveness of using Situat3DChange as a training dataset for MLLMs. The established dataset and source code are publicly available at: https://github.com/RuipingL/Situat3DChange.

## 1  Introduction

*"No man ever steps in the same river twice, for it is not the same river and he is not the same man."*

— Heraclitus of Ephesus

The physical world continuously evolves through both object transformations and shifting situational contexts, creating a dynamic interplay between environmental and human-centered changes. While private spaces are generally maintained in an orderly configuration, even minor positional shifts can transform familiar pathways into obstacles for visually impaired individuals [1, 2], with the restoration of previous arrangements [3, 4, 5, 6, 7] representing a key challenge for embodied agents. To facilitate effective human-AI collaboration in dynamic environments, it is essential for both humans and agents to develop shared mental maps and shared situational awareness [8] that enable them to jointly interpret and respond to evolving scenes.

---

[*] Corresponding author.

39th Conference on Neural Information Processing Systems (NeurIPS 2025) Track on Datasets and Benchmarks.

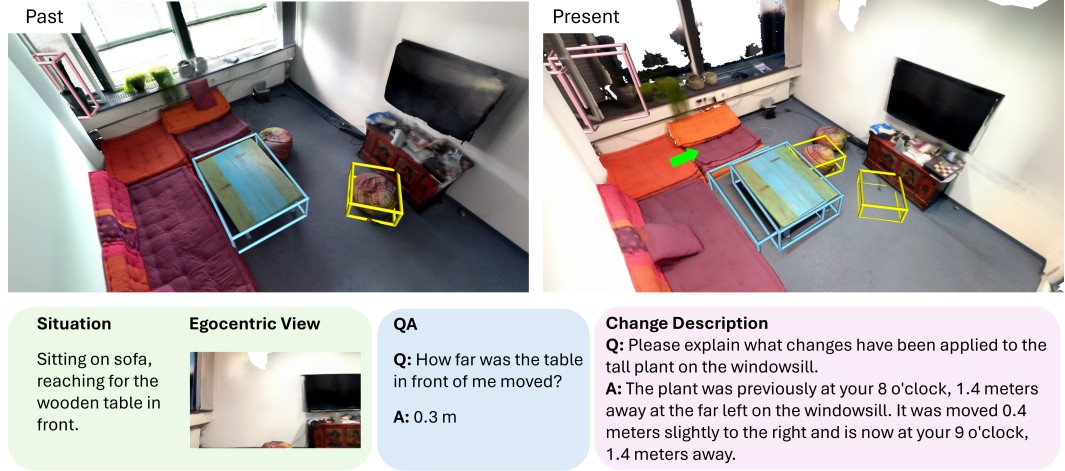

Figure 1: **Perception** for comprehensive understanding of the dynamic scene with situational awareness, including concise QA and change description.

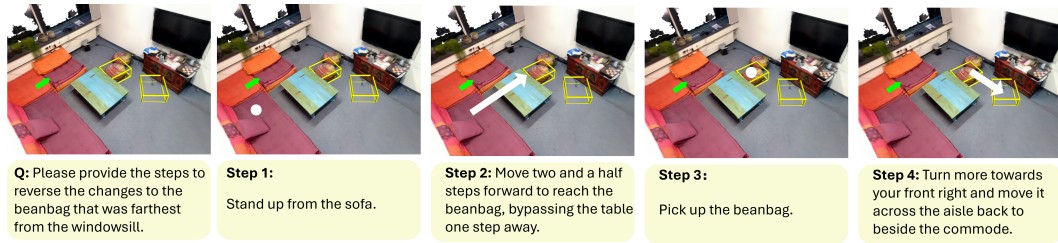

Figure 2: **Action** for rearrangement instructions to revert changes from the current situation.

Despite the dynamic nature of both situations and scenarios in the physical world, current research approaches address these aspects in isolation rather than holistically. Existing real-world 3D situated reasoning datasets, whether human-annotated [9] or LLM-generated [10, 11], typically rely on static scenarios that fail to capture environmental dynamics. Conversely, 3D change understanding datasets [12, 13, 14, 15, 16] often utilize synthetic data lacking situational context, missing the subtle, everyday changes characteristic of real environments. Although 3RScan [17] offers a real-world dataset for change detection, a comprehensive framework that integrates both situational awareness and physical transformations remains underexplored.

To explore 3D scene understanding with both dynamic scenes and situations simultaneously, we introduce **Situat3DChange**, a **situated 3D change** dataset comprising 903 real-world scan pairs yielding $174K$ paired data instances. This dataset frames environmental understanding through the perception-action model [18], which integrates question-answering (QA) and change description tasks (*i.e.*, perception, as in Fig. 1) with rearrangement instruction task (*i.e.*, action, as in Fig. 2), creating a foundation for AI systems that can both interpret and manipulate their surroundings with human-like awareness and adaptability. QA responses are intentionally brief and concise, whereas change descriptions and rearrangement instructions are long-form.

The development of our large-scale dataset raises fundamental questions about the cognitive alignment between AI and human perception. While the proliferation of LLM-generated training data based on bounding box centers [19, 20] and scene graphs [21, 22, 10, 23] has enabled seamless scaling while maintaining similarity to human natural language, this raises concerns: *(Q1) Do current LLM-based data generation methods produce content that reflects embodied shared mental models and situational awareness comparable to human cognition, or do they merely mimic linguistic patterns?* For situated 3D change understanding, *(Q2) can LLM effectively define and reference points of interest to anchor detected changes in ways that align with human perception?* For the first question, as shown in Fig. 3, across industry and engineering domains, the world is typically perceived in a Cartesian coordinate frame, generating bounding boxes and scene graphs. We interviewed 30 diverse individuals, including two blind individuals and native speakers of four languages, revealing perceptual discrepancies in relative spatial direction interpretation. Most perceive surroundings in a

cylindrical coordinate manner: when referencing left or right, a view direction is specified relative to a reference object [24], and the object closer to the observer is considered in front. This prevents agents and humans from sharing a mental map. Addressing our second question on LLM-generated data's ability to reference spatial changes, evidence shows a clear limitation. As demonstrated by 3DSSG [25], scene graphs lack perceptual sensitivity to detect critical 3D changes like object rotations or subtle positional shifts of approximately 10 cm. LLM-generated data, dependent on scene graph representations, fails to capture spatial changes that humans immediately deem significant.

To create our situation-aware change understanding dataset, we collected $11K$ human annotations from seven co-authors experienced in assisting visually impaired people. These annotators interpreted the labeled change of 3RScan in terms of reason, warning, description, and rearrangement instruction, and identified distinctive features for querying objects. We then enhanced this foundation by integrating egocentric and allocentric spatial information alongside object attributes using an LLM, enabling efficient scaling to fully situated data while preserving the human perceptual framework.

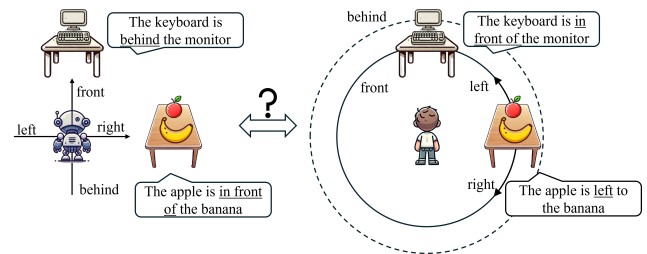

(a) Robotic cognition.  (b) Human cognition.

Figure 3: Different senses of relative spatial directions between robots and humans result in different mental maps.

As no existing 3D MLLMs process paired point clouds effectively, we introduce **SCReasoner**, a novel paradigm for situated 3D change understanding. Previous approaches [21, 10, 26] prepend all modality tokens to the language decoder input, which is inefficient for our case of highly similar point clouds, causing redundancy and failing to highlight changes. SCReasoner takes advantage of Mamba's selective nature [27] and parameter-less star operations [28] to focus on differences between point clouds with minimal parameter overhead. Experiments evaluating MLLMs on Situat3DChange tasks show that training with our dataset improves understanding of spatial change and transfers positively to other domains, confirming its value for developing perceptually aligned embodied agents and demonstrating beneficial scaling effects with increased data volume.

- We introduce Situat3DChange, a situated 3D change dataset with $121K$ QA pairs, $36K$ change descriptions, and $17K$ rearrangement instructions built on $11K$ human annotations for dynamic scenario understanding with situation awareness.

- To adapt to the Situat3DChange tasks, we introduce SCReasoner, a token-efficient MLLM architecture that compares scene pairs effectively.

- We conduct an extensive analysis of state-of-the-art MLLMs on each task, highlighting both the limitations and strengths of MLLMs on the Situat3DChange task. We further analyze the scaling effects of Situat3DChange and cross-domain transferability.

## 2  Related Work

**Situated scene understanding.**  Beyond allocentric scene understanding [24, 29, 30, 31], structured egocentric scene understanding has been explored. Egocentric Scene Graphs [32, 33, 34, 35] capture interactions and spatial relations from a first-person view. Ego4D [36] supports understanding of daily activities from egocentric videos, yet lacking full 3D reconstruction. Besides, 3D Scene Graph [37] provides hierarchical representations of indoor environments with known camera poses. LSceneLLM [38] enables cross-room spatial reasoning. For situated reasoning and QA, most prior work focused on simulated environments [39, 40, 41, 42]. SQA3D [9] combines 3D scenes with crowdsourced QA, while MSQA [10] and Spartun3D [11] scale this using LLM-generated questions. SceneVerse [22] and 3DLLM [19] support embodied reasoning in real-world contexts. Embodied agents [21, 43] are trained with shared situational awareness. For robotic applications, RoboSpatial [44] is a dataset that focuses on spatial context, object compatibility, and configuration. Phys100K [45] is a large-scale multi-robot QA dataset. Thinking in Space [46] introduces a dataset designed to facilitate spatial understanding based on short egocentric videos. 3D-GRAND [47] is a large-scale, densely grounded 3D dataset aimed at mitigating hallucinations. M3DBench [48]

Table 1. Comparison of 2D/3D scene understanding datasets. Allo.: allocentric; Ego.: egocentric.

| Dataset | Change | Perspective | Data Type | Scenes | Annotation | QA | | Long-form Text | |
|---|---|---|---|---|---|---|---|---|---|
| | | | | | | #types | #pairs | #description | #rearrangement |
| Dy2Change [15] | ✓ | allo. | 3D sim | 38K pairs | Template | - | - | 661K | - |
| EmbSCU [61] | ✓ | allo. | 2D sim | 20.5K pairs | Template | - | - | 21K | 21K |
| PanoSCU [65] | ✓ | allo. | 2D sim | 2650 pairs | Template | - | - | 9K | 9K |
| ReferIt3D [24] | ✗ | allo. | 3D scan | 707 scenes | Human | - | - | 41.5K | - |
| ScanRefer [29] | ✗ | allo. | 3D scan | 800 scenes | Human | - | - | 51.6K | - |
| 3D-GRAND [47] | ✗ | allo. | 3D sim | 40K scans | Template+GPT | 6 | 5.5M | 118K | - |
| MMScan [49] | ✗ | allo. | 3D scan | 5.2K scans | Template+GPT+Human | 5 | 1.76M | 4.06M | - |
| OpenEQA [71] | ✗ | ego. | 3D scan | 180 scans | Human | 7 | 1.6K | - | - |
| SQA3D [9] | ✗ | ego. | 3D scan | 650 scans | Human | 6 | 33.4K | - | - |
| MSQA [10] | ✗ | ego. | 3D scan | 1734 scan | GPT | 9 | 251K | - | - |
| VSI-Bench [46] | ✗ | ego.+allo. | 2D video | 288 videos | Human | 8 | 5K | - | - |
| RoboSpatial [44] | ✗ | ego.+allo. | 3D scan | 5223 scans | Template | 3 | 3M | - | - |
| Situat3DChange (Ours) | ✓ | ego.+allo. | 3D scan | 903 pairs | Human+GPT | 9 | 121K | 36K | 17K |

and MMScan [49] address situated scene understanding from global to local perspectives, but are limited in terms of embodied agent viewpoints and static scenarios. SpatialLLM [50] not only addresses spatial reasoning from a human perspective but also models object orientation relationships. In accessibility, VizWiz [51, 52, 53] focuses on images taken by blind users, while GuideDog [54] provides outdoor egocentric data for navigation. SANPO [55] adds multiview and multi-sensor scene understanding. Yet, existing datasets are limited to static scenes. In contrast to previous situational scene understanding datasets, we introduce Situat3DChange for 3D scene understanding of dynamic scenarios and situations.

**Spatial change understanding.** Most research on change understanding focuses on 2D images [56, 57, 58]. Factors affecting change blindness have been studied by Martin *et al.* [59] and Ma *et al.* [60]. Some methods assume fixed viewpoints [61, 13, 62], whereas inpainting techniques simulate visual changes [63, 64]. Besides, PanoSCU [65] uses synthetic panoramas for change captioning. STVchrono [66] explores continuous temporal change recognition. In 3D environments, Dy2Change [15] and ChangeSim [16] emulate human-like dynamic scanning, but neither performs situation-aware comparisons. Beyond textual descriptions, 3RScan [17] provides real-world data for object relocalization, and 3DSSG [25] augments it with scene-graph annotations of object attributes. For street-scene point clouds, efforts include camera relocalization [67] and change localization [68]. To detect multiple simultaneous changes, graph-based [69, 70] and transformer-based [13] approaches have been explored, yet they remain largely task-specific and confined to synthetic environments. In contrast, we introduce a general-purpose 3D MLLM paradigm capable of robustly understanding multiple changes in complex, real-world scenes. We compare Situat3DChange with embodied scene and change understanding datasets in Tab. 1.

## 3 Situat3DChange Dataset

In Sec. 3.1, we state the details of data generation, including: situation sampling, long-form text generation, query generation, QA generation, and data quality control. In Sec. 3.2, we show the statistics of the dataset and the metrics of our benchmark.

### 3.1 Data Generation

**Situation sampling.** We follow MSQA [10] to sample the position and orientation in three categories: sitting, standing, and interacting with objects. To enhance the variety of the situations, each situation refers to a unique instance object in the scene to construct a concise instance-aware situation description, *e.g.*, *sitting on sofa_22*. Besides the situation description [9, 10], we also consider the approximate eye height ($157\pm10$ cm for standing and $76.5\pm5$cm fir sitting) [72] and the tilt of the head ($\pm30$ degrees) [73] to capture the egocentric view and the panorama to facilitate the understanding of the situation. The concise situation will then be expanded with comprehensive spatial information to the descriptive situation with at least two reference objects, *e.g.*, *sitting on sofa, reaching for the wooden table in front.*, as shown in Fig. 1.

**Long-form text generation.** Regarding how information should be delivered [1, 74, 75, 76], a combination of egocentric and allocentric perspectives, along with categorical and coordinate in-

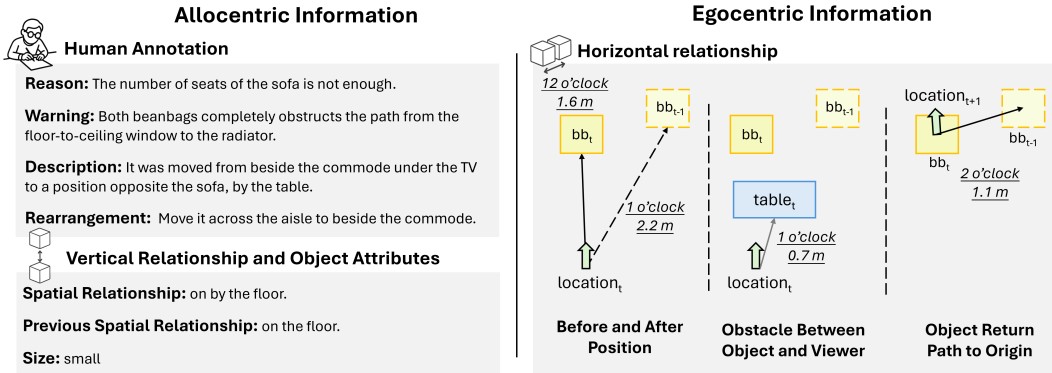

Figure 4: Allocentric and egocentric information used for generating situated QA pairs, change descriptions, and rearrangement instructions.

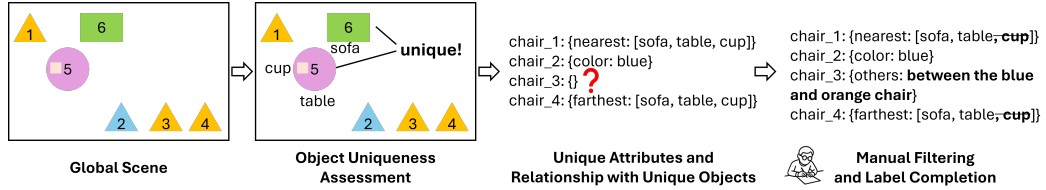

Figure 5: Distinctive features used to refer to objects for generating queries about change descriptions and rearrangement instructions. △, ◯, □, and ▢ refer to chairs, table, cup, and sofa.

formation, supports the human perception-action loop in 3D environments. To facilitate human-AI collaboration, the human sense of mental mapping and change reference should be introduced.

Seven co-authors, all with experience collaborating with blind users, annotated every object that was labeled as a change in 3RScan [17]. For each object they supplied four fields: (i) a plausible **Reason** for the change, (ii) **Warning** information when the change could create an obstacle along a typical path, (iii) a detailed change **Description** that captures the object's rotational and translational displacement relative to neighbouring items, and (iv) **Rearrangement** instructions explaining how to restore the object to its original pose. While these annotations provide allocentric horizontal cues, cylindrical-coordinate data are still required to reflect humans' viewpoint, *i.e.*, egocentric horizontal and allocentric vertical information. Building on 3DSSG [25], we extract vertical relationships and object attributes, compute the previous and current egocentric positions of moved objects and any obstacles they create, and finally determine the direction and straight-line distance a user must travel from their current orientation to reach the displaced object and then return it to its original location, as shown in Fig. 4. The raw data are organized into a JSON file that serves as input to GPT-4 [77], a model shown to produce highly human-like text [78], for generating situated change descriptions and rearrangement instructions. For change descriptions, we follow a clockwise ordering and express distances in *meters*, as in [79]. For rearrangement instructions, exploration during navigation is preferred over step-by-step guidance [80], while using *steps* as a unit is generally easier to understand than *meters* [81].

**Query generation for long-form text.** Prior works [13, 14, 61] in change captioning rely on simulators, where data quality is tightly controlled and every change is deterministically specified. Work that does address multiple changes [13] concatenates all descriptions into one sentence sequence. That strategy is unsuitable for our real-world dataset, where the data are noisier and many subtle changes remain unlabeled. Inspired by the object identification task [29, 24], we propose a pipeline that generates queries uniquely identifying each object through its distinctive features, as in Fig. 5.

First, any object that is already unique in the scene is promoted to a landmark because it can be referenced directly, as in queries like *"What change happened to the table?"* for change description for *table_5*. Next, we extract three candidate distinctive features for every remaining object automatically: (i) its distinctive color, (ii) its horizontal extremity, whether it is nearest to or farthest from a landmark, and (iii) its vertical spatial relation to the landmarks, *e.g. "What alterations occurred to*

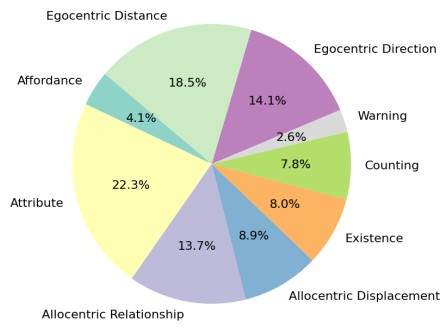

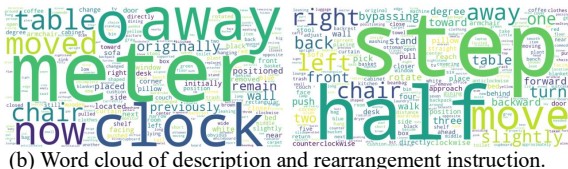

(b) Word cloud of description and rearrangement instruction.

| Subtask | Number of Pairs | | Avg. Word Length | |
|---|---|---|---|---|
| | Train | Validation | Query | Answer |
| QA | 106,317 | 14,713 | 7.82 (2.00) | 2.18 (1.16) |
| Description | 30,963 | 4,729 | 11.20 (3.43) | 29.48 (10.63) |
| Rearrangement | 14,832 | 2,254 | 13.54 (3.97) | 29.62 (9.02) |

(a) Distribution of QA types

(c) Distribution of query and answer counts across subsets, with average word length analysis.

Figure 6: Statistics of the Situat3DChange dataset: (a) distribution of QA types, (b) word clouds for change descriptions and rearrangement instructions, and (c) data-split sizes with word-length distributions for each task. The numbers in parentheses indicate standard deviations.

*the chair nearest to the table?"* for *chair_1*. Finally, the co-authors review these features, discarding ones that depend on landmarks unlikely to be recognized (*e.g.*, a cup) and manually adding an extra feature whenever an object still lacks a uniquely identifiable feature, *"What change has been made to the chair between the blue and the orange chair?"* for *chair_3*. Each long-form query for a changed object centers on a single identifiable feature from the current or prior scene.

**QA generation.** Our dataset introduces nine specialized QA types for 3D situated change understanding. For egocentric understanding, **Egocentric Distance (Pre/Post)** and **Egocentric Direction (Pre/Post)** questions focus on the object's previous and current locations relative to the person's position and orientation, while **Warning** questions detect when a moved object becomes a potential obstacle from the person's perspective. For allocentric understanding, **Allocentric Displacement** captures how far an object has moved, and **Allocentric Relationship** assesses how its spatial relationship with other objects has changed. To support general scene understanding, we include **Affordance**, **Attribute**, **Existence**, and **Counting** questions, which cover functional, descriptive, and numerical aspects of the scene before and after the change.

We use nearly the same raw data as in the long-form text-generation setting, but we exclude the human annotations. The Object-centric Chain-of-Thought (O-CoT) [21] is employed to generate the dataset, conditioned on the target object's label and index per QA instance.

**Data quality control.** Long-form texts originate from human annotations provided by seven co-authors experienced in working with blind individuals. While these annotations do not require separate human verification, they are automatically expanded to include egocentric distances, directions, and object attributes. QA pairs are generated directly from the raw ground-truth labels and indexed using O-CoT. Since each sentence is tagged with the object index and QA type, we can automatically verify its correctness by cross-referencing it with the original data for dataset generation, while excluding incorrect and indeterminate answers.

## 3.2 Data Statistics and Metrics

**Data statistics.** Since 3RScan [17] lacks publicly available test-set labels and is widely adopted as a benchmark for MLLM training [21, 11, 29], where its validation set is used for model selection, we do not further divide the validation set into validation and test splits, unlike SQA3D [9]. Because our task is specific and meant for the MLLM fine-tuning stage, we keep the original training/validation split and use it as a hold-out evaluation setup for MLLMs. Fig. 6 (a) shows a balanced distribution of question–answer types: 35.2% are egocentric spatial questions (Distance, Direction, Warning), 22.6% are allocentric spatial (Relationship, Displacement), and 42.2% concern general visual information (Affordance, Attribute, Existence, Counting). Fig. 6 (b) visualizes a word cloud of change descriptions and rearrangement instructions. Fig. 6 (c) reports the number of training and validation examples for each task, together with the average word length of queries and answers. Additional scene- and situation-level statistics are provided in the App. B.

**Metrics.** To evaluate MLLM-predicted long-form change descriptions and rearrangement instructions, we use reference-based metrics, CIDEr, METEOR, ROUGE, and BLEU-4 as well as sentence similarity scores computed using a BERT-based model [82] and GPT-based evaluation metrics [71].

The ground truth answers for QA types other than distance and displacement are brief, but the LLM-generated responses are usually more descriptive. Following MSQA [10], we adopt a GPT-based evaluation metric with fine-grained prompts to assess LLM-generated responses uniformly:

$$C = \frac{1}{N} \sum_{i=1}^{N} \frac{s_i - 1}{4} \times 100\%, \tag{1}$$

where $C$ denotes the correctness score over $N$ samples, and $s_i$ (ranging from 1 to 5, with higher scores indicating better quality) is the rating generated by GPT when given the question, ground-truth answer, and model response. The GPT model used for evaluation is fixed to a specific timestamp to ensure consistency across all scores. The strong correlation between human and GPT scoring is confirmed in App. D.2.

For accurate distance evaluation, we adopt a different metric. In egocentric distance QA, we measure the horizontal displacement from the subject's standing point to the object, rather than the eye-to-object distance, which introduces errors due to variations in human height. For allocentric displacement, which also accounts for any movement of the subject, a stationary object results in $d_{\text{gt}} = 0$. In such cases, even a small prediction error (e.g., $d_{\text{pred}} = 0.1\,\text{m}$) causes the conventional depth-estimation metric $\text{REL} = \frac{|d_{\text{pred}} - d_{\text{gt}}|}{d_{\text{gt}}}$ to diverge due to division by zero. These infinite errors are often triggered by minor inaccuracies in bounding box annotations or subject position. Other standard depth scores share the same limitation for our scenario, so we propose a revised REL:

$$\text{score} = \begin{cases} 1, & \text{if } d_{\text{gt}} = 0 \text{ and } d_{\text{pred}} = 0, \\ 0, & \text{if } d_{\text{gt}} = 0 \text{ and } d_{\text{pred}} \neq 0, \\ 1 - \min\left(1, \frac{|d_{\text{pred}} - d_{\text{gt}}|}{d_{\text{gt}}}\right), & \text{if } d_{\text{gt}} \neq 0 \text{ and } d_{\text{pred}} \neq 0. \end{cases} \tag{2}$$

The score range matches the GPT-score (higher is better), and echoes binocular disparity, which tolerates greater error at longer distances. $d_{gt}$ and $d_{pred}$ denote the ground truth and the predicted distance of egocentric distance and allocentric displacement.

## 4 MLLMs for Situated 3D Change Understanding

**3D MLLM: SCReasoner.** As there is currently no existing 3D MLLM that takes two point clouds as input, we propose a new approach tailored to the Situat3DChange task where the two point clouds are similar with only minor changes. Previous methods [10, 21] typically prepend all modality tokens together at the input of the language decoder, which is redundant in this case: similar tokens from both point clouds overwhelm the meaningful differences, and the human question is not always related to the previous scene. To address this, we introduce SCReasoner, a new paradigm in which both point clouds share a common encoder to embed them into tokens. The informative tokens from the previous scene are selected and fused with tokens from the current scene. For token selection, we explore the selective nature of Mamba [27], while for token fusion, we consider the star operation (element-wise multiplication) [28], which has been shown to effectively map inputs into high-dimensional representations. Our architecture is built on top of the flexible and widely adopted LEO framework [21]. Despite taking two point clouds as input, our design introduces only a small number of additional parameters for token selection. The other input modalities remain the same as in the situated task setup of LEO.

For comparison, we re-implemented LEO by prepending tokens from the two point clouds at the input, consistent with how tokens from other modalities are handled. SCReasoner instead introduces a selective comparison projector before the decoder to enhance token efficiency, consisting of a projection module for focusing on scene changes and a fusion module for removing redundancy. We tested a linear layer and Mamba for projection, and element-wise addition (+) or multiplication (*) for parameter-free fusion.

Table 2. Results on the long-form text tasks, including change description and rearrangement instruction. *"mamba"* denotes using the Mamba module to select features from the previous scene, while *"linear"* refers to a single linear layer for dimensionality projection. *"+"* and *"*"* indicate the parameter-free fusion operations, namely element-wise addition and star operation, respectively.

| Method | Setting | Change Description | | | | | | Rearrangement Instruction | | | | | |
|---|---|---|---|---|---|---|---|---|---|---|---|---|---|
| | | CIDEr | BLUE-4 | METEOR | ROUGE | sim | GPT | CIDEr | BLUE-4 | METEOR | ROUGE | sim | GPT |
| DeepSeek-VL-7B [86] | zero-shot | 0.4 | 0.6 | 10.1 | 11.2 | 50.3 | 2.3 | 0.0 | 0.4 | 11.0 | 7.9 | 51.8 | 0.0 |
| DeepSeek-VL2-1.3B [87] | zero-shot | 6.6 | 1.3 | 7.3 | 14.0 | 53.1 | 2.3 | 2.1 | 0.7 | 10.9 | 12.2 | 53.0 | 0.3 |
| Janus-7B [88] | zero-shot | 6.0 | 0.1 | 9.4 | 15.9 | 49.3 | 1.7 | 0.4 | 1.3 | 16.0 | 14.2 | 55.7 | 0.2 |
| Qwen2.5-VL-7B [89] | zero-shot | 1.4 | 1.1 | 11.4 | 13.5 | 52.9 | 5.5 | 6.7 | 1.3 | 11.0 | 15.0 | 56.6 | 0.4 |
| LLava-NEXT-7B [90] | zero-shot | 0.1 | 0.6 | 10.5 | 11.0 | 49.9 | 2.6 | 0.0 | 0.4 | 10.8 | 7.4 | 53.3 | 0.0 |
| InternVL2-7B [84] | zero-shot | 0.2 | 0.2 | 7.7 | 6.5 | 46.1 | 0.9 | 0.2 | 0.2 | 7.7 | 6.5 | 46.1 | 0.0 |
| DeepSeek-VL-7B [86] | one-shot | 14.0 | 3.9 | 13.3 | 24.6 | 66.2 | 2.8 | 7.2 | 7.4 | 23.1 | 28.5 | 68.3 | 3.4 |
| DeepSeek-VL2-1.3B [87] | one-shot | 7.4 | 1.8 | 8.3 | 15.3 | 56.7 | 2.5 | 2.1 | 1.2 | 13.2 | 13.8 | 56.0 | 0.7 |
| Janus-7B [88] | one-shot | 14.4 | 4.2 | 14.7 | 23.4 | 65.2 | 2.7 | 12.9 | 10.3 | 26.7 | 32.7 | 73.2 | 4.7 |
| Qwen2.5-VL-7B [89] | one-shot | 1.4 | 3.7 | 11.8 | 23.1 | 62.3 | 2.7 | 20.4 | 4.7 | 19.1 | 24.1 | 70.0 | 4.9 |
| LLava-NEXT-7B [90] | one-shot | 10.4 | 3.6 | 19.2 | 23.5 | 65.0 | 3.4 | 0.0 | 1.4 | 16.7 | 11.9 | 61.8 | 0.4 |
| InternVL2-7B [84] | one-shot | 5.6 | 1.5 | 13.4 | 17.3 | 59.0 | 3.8 | 5.6 | 1.5 | 13.4 | 17.3 | 59.0 | 3.7 |
| InternVL2-7B [84] | fine-tuning | 36.5 | 11.7 | 24.1 | 36.2 | 73.5 | 8.2 | 56.1 | 17.5 | 26.4 | 40.0 | 79.5 | 16.3 |
| LEO [21] | fine-tuning | 45.9 | 14.0 | 26.3 | 42.1 | 74.6 | 12.7 | **73.0** | **19.0** | 27.0 | 41.1 | 81.7 | 30.1 |
| SCReasoner (linear+) | fine-tuning | 51.2 | 14.6 | 26.9 | 42.5 | 75.8 | 12.6 | 72.4 | 18.6 | 27.1 | 41.2 | 81.9 | 30.3 |
| SCReasoner (linear*) | fine-tuning | 50.1 | 14.4 | 26.8 | 42.4 | 75.6 | 13.4 | 72.9 | 18.5 | **27.2** | 41.4 | **82.0** | 30.3 |
| SCReasoner (mamba+) | fine-tuning | 52.9 | 15.0 | 27.2 | 42.9 | 76.0 | 13.3 | 71.7 | 18.1 | 26.9 | 40.7 | 81.7 | 30.5 |
| SCReasoner (mamba*) | fine-tuning | **53.6** | **15.2** | **27.2** | **43.0** | **76.1** | **13.9** | 72.9 | 18.3 | **27.2** | 40.9 | 81.9 | **30.7** |

**2D MLLMs baselines.** We evaluate state-of-the-art open-source models on MMMU [83] using both zero-shot and one-shot settings, and fine-tune the best-performing model among them on MMMU, InternVL2 [84], on our dataset. Except for DeepSeek-VL2, which uses a large Mixture-of-Experts (MoE) architecture, we select the 7B versions of all models, consistent with LEO. Since our scenes are small, we use panoramas, which can reconstruct the surrounding 3D scene and represent the egocentric view [85], as input to 2D MLLMs. One panorama is captured from a random point in the previous scene (the origin of the world coordinate), and the other from the position and orientation of the current situation. The panoramas are generated using the method of [85], rendered from a six-view skybox.

Since the performances of MLLMs depend not only on their architecture but also on the training data, we conduct a second-stage fine-tuning on their existing weights for 5 epochs using a combined set of all three tasks, and hold-out validation. Further details are in the App. C.

## 5 Experiments

**Results on long-form tasks.** As presented in Tab. 2, the long-form tasks remain challenging for zero- and one-shot MLLMs. With the inclusion of a single example, model performance improves but remains suboptimal. Among these models, Qwen2.5 achieves the best results. After fine-tuning, InternVL shows a significant improvement in overall scores, highlighting the importance of modeling scene changes during the fine-tuning of MLLMs for scene understanding. 3D MLLMs outperform their 2D counterparts. Our SCReasoner outperforms InternVL2 with gains of 5.7% and 15.4% on change description and rearrangement instruction tasks, respectively, based on GPT-based evaluation. Compared to the baseline LEO, SCReasoner achieves 1.2% and 0.6% improvements on the same tasks. Both the use of Mamba for token selection and the star operation for token fusion contribute to improved scene change understanding, highlighting the effectiveness of focusing on informative tokens for point cloud comparison.

**Results on concise QA task.** Tab. 3 shows QA results on the Situat3DChange (S3C) dataset. Among one-shot 2D MLLMs, DeepSeek-VL performs best, followed by LLaVA-Next. Comparing fine-tuned 3D and 2D MLLMs, InternVL-2 slightly outperforms SCReasoner by +0.2% on average. However, 3D MLLMs are better at allocentric understanding, while the 2D MLLM excels in egocentric tasks such as Distance and Direction. This suggests that S3C fine-tuning effectively calibrates 2D MLLM for panoramic image understanding. The use of point clouds enables more holistic scene perception. This also explains why InternVL performs poorly on long-form answers, as panoramas lack a comprehensive allocentric context.

**Scaling effect.** We explore the scaling effects of SCReasoner on each task of S3C using different training data scales, evaluated on the full validation set. Following the methodology of MSQA [10], we investigate three key factors influencing scaling: (1) Sample: randomly downsampling query-answer pairs across all tasks; (2) Situation: randomly downsampling situations, which removes

Table 3. Results on QA task. *"pano"* and *"pcd"* denote panoramas and point clouds.

| Models | Input | Setting | Affordance | Attribute | Existance | Counting | Warning | Allo. Rel. | Allo. Dis. | Ego. Dir. | Ego. Dis. | Average |
|---|---|---|---|---|---|---|---|---|---|---|---|---|
| DeepSeek-VL-7B [86] | pano | one-shot | 36.0 | 47.4 | 60.8 | 35.5 | 15.9 | 27.1 | 41.2 | 29.8 | 37.3 | 36.8 |
| DeepSeek-VL2-1.3B [87] | pano | one-shot | 11.9 | 45.8 | 18.2 | 30.7 | 19.3 | 26.2 | 20.7 | 21.2 | 42.9 | 26.3 |
| Qwen2.5-VL [89] | pano | one-shot | 36.2 | 57.3 | 21.4 | 24.8 | 19.5 | 48.7 | 33.6 | 29.0 | 46.9 | 35.3 |
| LLaVA-OV-7B [91] | pano | one-shot | 46.3 | 49.0 | 47.3 | 22.6 | 19.1 | 38.2 | 16.2 | 21.4 | 20.6 | 31.2 |
| LLaVA-NEXT-7B [90] | pano | one-shot | 42.7 | 49.3 | 61.2 | 33.1 | 26.5 | 27.4 | 41.2 | 26.0 | 20.5 | 36.4 |
| InternVL2-7B [84] | pano | one-shot | 58.5 | 41.4 | 58.3 | 27.3 | 15.4 | 31.7 | 28.5 | 23.7 | 36.8 | 35.7 |
| InternVL2-7B [84] | pano | fine-tuning | 64.3 | **60.6** | 69.7 | 37.5 | 42.0 | 58.0 | 40.9 | **56.1** | **57.3** | **54.0** |
| LEO [21] | pcd | fine-tuning | 70.0 | 57.1 | **72.9** | 40.6 | 46.1 | 50.2 | 40.5 | 38.7 | 51.1 | 51.9 |
| SCReasoner(linear*) | pcd | fine-tuning | **70.3** | 56.0 | 72.4 | **41.0** | 46.2 | 61.7 | 41.9 | 40.2 | 50.7 | 53.4 |
| SCReasoner(mamba*) | pcd | fine-tuning | 68.9 | 60.0 | 71.4 | 40.8 | 44.0 | **63.5** | **43.5** | 43.3 | 51.4 | 53.8 |

Figure 7: Effects of scaling training data on three tasks, based on the same full validation set.

all associated QA pairs; and (3) Scan Pair: randomly downsampling scan pairs, along with their related QA pairs. As shown in Fig. 7, we observe a consistent trend of improvement when scaling up along the three factors in the QA and change description tasks. The inconsistent scaling effect observed in the rearrangement instruction task may be due to the use of coarse spatial references, such as directions (left and right) and distances (steps), instead of the more precise annotations (*e.g.*, clockwise direction and meters) used in the QA and change description tasks. These coarse spatial concepts were already seen during the initial LEO training on the 3RScan benchmark. Moreover, the relatively small amount of data for rearrangement instructions may cause results to be affected by the stochasticity of data selection.

**Cross-domain transfer.** In Tab. 4, we evaluate models fine-tuned on different domains using ScanNet-based benchmarks: Scan2Cap [30], ScanQA [31], and SQA3D [9], all of which were seen during LEO's instruction-tuning. Fine-tuning only on our S3C dataset leads to catastrophic forgetting on these benchmarks. In contrast, fine-tuning on the combined ScanNet benchmarks (SN) improves performance on Scan2Cap and SQA3D while maintaining accuracy on ScanQA. Adding S3C to SN further improves results across all benchmarks, likely due to S3C's human-authored language, which enhances model generalization compared to LEO's LLM-generated data.

In Tab. 5, we present the domain transfer results on Situat3DChange. The average score of SCReasoner trained on both SN and S3C is slightly lower ($-1.3\%$) than SCReasoner trained solely on S3C. One reason is the significantly dropped performance ($-14.4\%$) on the egocentric direction task. We attribute this to SN expressing egocentric direction with *left* and *right* instead of precise clockwise terms, which may confuse both the model and the GPT-based evaluator. Training on both datasets yields comparable or improved performance on QA and instruction tasks, and significantly boosts change description by enhancing dynamic scene understanding.

Table 5. Cross-domain performance on the Situat3DChange (S3C) dataset. *"Desc."* denotes change descriptions, and *"Rearr."* denotes rearrangement instructions.

| 2nd FT | QA | | | | | | | | | | Desc. | Rearr. |
|---|---|---|---|---|---|---|---|---|---|---|---|---|
| | Affordance | Attribute | Existance | Counting | Warning | Allo. Rel. | Allo. Dis. | Ego. Dir. | Ego. Dis. | Average | | |
| S3C | 68.9 | **60.0** | 71.4 | 40.8 | 44.0 | **63.5** | **43.5** | **43.3** | **51.4** | **53.8** | 13.9 | **30.7** |
| S3C+SN | **69.4** | 56.5 | **73.3** | **42.6** | **46.7** | 61.6 | 43.0 | 28.9 | 50.2 | 52.5 | **16.8** | 30.1 |

# 6 Conclusion

In this work, we introduce Situat3DChange, a large-scale situated change understanding dataset featuring three tasks: question answering ($121K$), change description ($36K$), and rearrangement instruction ($17K$). To facilitate human-AI collaboration with shared mental maps, we incorporate $11K$ human-sensed changes, coupling allocentric and egocentric relationships with object attributes to generate the dataset. To enhance understanding of dynamic scenes and situations, we propose SCReasoner, a token-efficient MLLM paradigm. Our comprehensive experiments highlight

Table 4. Cross-domain effect on the union of ScanNet benchmarks (SN). Results in parentheses follow the refined exact-match protocol from LEO [21].

| Method | 2nd FT | Scan2Cap (val) | | | | | ScanQA (val) | | | | | SQA3D (test) |
|---|---|---|---|---|---|---|---|---|---|---|---|---|
| | | C | B-4 | M | R | Sim | C | B-4 | M | R | EM@1 | EM@1 |
| SCReasoner (LEO) | - | 72.4 | 38.2 | 27.9 | 58.1 | 55.3 | 101.4 | 13.2 | 20.0 | 49.2 | 24.5 (47.6) | 50.0 (52.4) |
| SCReasoner | S3C | 14.5 | 14.8 | 20.0 | 50.1 | 40.7 | 54.2 | 1.5 | 9.4 | 28.5 | 15.4 (31.4) | 37.7 (40.6) |
| SCReasoner | SN | 82.2 | 44.1 | 30.7 | 65.6 | 64.4 | 103.1 | 15.2 | 20.1 | 48.5 | 23.4 (46.7) | 51.5 (**54.2**) |
| SCReasoner | S3C+SN | **83.4** | **44.6** | **31.0** | **65.8** | **64.9** | **104.3** | **15.7** | **20.3** | **49.3** | **24.6** (**48.0**) | **51.8** (**54.2**) |

the value of our Situat3DChange dataset and the effectiveness of SCReasoner. We hope this work will advance human-AI collaboration in dynamic environments, enabling more adaptive interactions across diverse and changing circumstances.

**Limitations and future work.** Despite the contributions, several limitations remain to be addressed in future work. The human change annotations are still limited, with each change annotated by only one annotator, which restricts diversity and limits the dataset's extensibility. Furthermore, the human-annotated data is not adapted to other domains, making the data generation pipeline difficult to transfer to other datasets. As the 3RScan test data is unavailable, Situat3DChange supports only hold-out evaluation for MLLM, limiting its use for task-specific VQA. This can be mitigated by generating pseudo-semantic annotations for the test set.

# Acknowledgment

We thank Weicheng Dai, Xinkai Qin, Yizhe Lu, and Haoxiang Liu for joining the human evaluation. This work was supported in part by the Ministry of Science, Research and the Arts of Baden-Württemberg (MWK) through the Cooperative Graduate School Accessibility through AI-based Assistive Technology (KATE) under Grant BW6-03, in part by funding from the pilot program Core-Informatics of the Helmholtz Association (HGF), in part by Karlsruhe House of Young Scientists (KHYS), and in part by the Helmholtz Association Initiative and Networking Fund on the HAICORE@KIT and HOREKA@KIT partition. This project is also supported in part by the National Natural Science Foundation of China under Grant No. 62473139, in part by the Hunan Provincial Research and Development Project (Grant No. 2025QK3019), and in part by the Open Research Project of the State Key Laboratory of Industrial Control Technology, China (Grant No. ICT2025B20).

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

# A  Data Generation

The data generation process includes situation sampling, long-form text generation, query generation for the long-form text, and QA generation. It is based on human observations of changes, object attributes, and allocentric object relationships in 3DSSG [25], as well as egocentric relationships between the human and the objects.

## A.1  Situation Sampling

We follow the situation categories of MSQA [10], namely sitting, interacting, and standing, but with more detailed geometric analysis:

**Sitting.** The 28 seat categories in 3RScan [17] are grouped into four types: 3 large seats with backrests (*e.g.*, sofa), 16 small seats with backrests (*e.g.*, armchair), 1 large seat without a backrest (bed), and 8 small seats without backrests (*e.g.*, beanbag). Seatable and backrest areas are classified by surface normals, or by nearby walls within $0.5$ m if no backrest exists. For small seats, the seating point is the bounding box center, oriented away from the backrest. For large seats, we select a point with a backrest behind and open space ($0.5$–$1$ m) in front. If no backrest or wall is present, the seat center is used, facing the room center.

**Interacting.** We consider objects with a dominant horizontal normal as interactable. A point is randomly selected from the standable floor regions, oriented toward the object center and within 5 degrees of the object's dominant normal, at a distance of $0.3$ to $0.5$ m from its bounding box.

**Standing.** The standing situation is anchored to the nearest object. In scenes with few objects (*e.g.*, a stairwell), multiple situations may share the same anchor object but differ in their proximity orientations, such as 3 o'clock (right) and 5 o'clock (back).

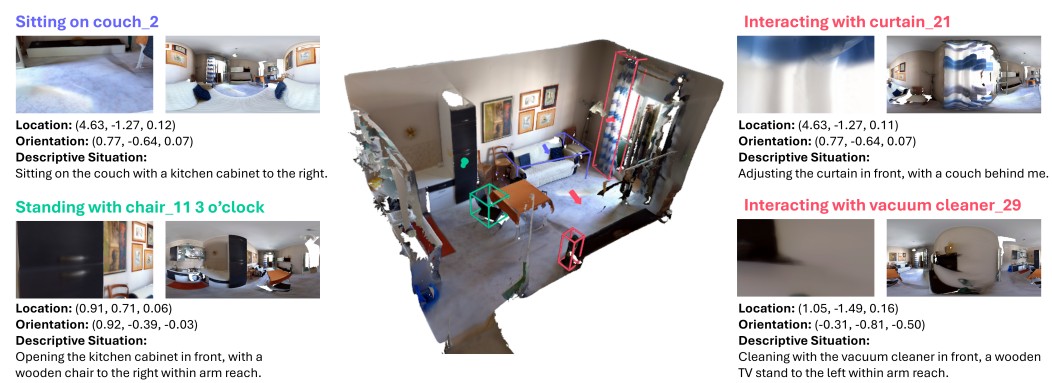

Figure 8: Examples of sitting, standing, and interacting situations. Each includes the location, orientation, egocentric view, panorama, and descriptive situations based on the scene's holistic context.

The aforementioned process generates a brief situation with an anchor object, location, and orientation, as shown in Fig. 8. The egocentric view and panorama are captured to represent the perspective of a wearable device or embodied agent. Since situations anchored solely to an object lack human-centered context and informativeness, we extend them into descriptive situations based on holistic scene information, incorporating at least two reference objects, as illustrated in Fig. 9.

system_prompt = ("You are an AI visual assistant tasked with expanding brief situational descriptions into 5 different detailed situation descriptions with human-object interactions within a 3D scene. Initially, the situation involves only one reference object, but your description should include at least two interacting objects. Exclude non-present objects. Each detailed description should be less than 20 words. The response should be in the format with 'S' is the detailed description and 'O' is the reference objects. Mention the directions (left, right, front, back) of all reference objects when standing. 'Interacting' should be an action conducted while standing, with the interacted object in front. Don't assume 'interacting' to be 'sitting'." )

data_sit = {"windowsill_4": {"attributes": ["metal", "dark", "gray"], "location": "left"}, "plant_7": {"attributes": ["tall"], "location": "left"}, "plant_8": {"location": "left, within arm reach"}, "beanbag_17": {"location": "left"}, "table_19": {"attributes": ["wooden", "blue", "green", "rectangular", "low", "narrow"], "location": "front, within arm reach"}, "cushion_20": {"location": "left, within arm reach"}, "cushion_21": {"attributes": ["tall", "wide"], "location": "left"}, "sofa_22": {"attributes": ["padded", "L-shaped", "orange", "pink", "wide"],"location": "below"}, "tv_24": {"attributes": ["black"], "location": "front, far away"}... }

data_interact = {"sink_7": {"attributes": ["white"], "location": "front, within arm reach"}, "mirror_9": {"location": "front, within arm reach"}, "toilet_13": {"attributes": ["seat down", "white", "tall", "wide"], "location": "left, within arm reach"}, "bucket_14": {"location": "back, within arm reach"}, "trash can_16": {"location": "front, within arm reach"}... }

data_stand = { "kitchen counter_2": { "attributes": ["stone", "rectangular", "white", "low"], "location": "front, within arm reach"}, "clutter_9": "location": "front, within arm reach", "clutter_11": {"location": "front, within arm reach"}, "window_13": {"attributes": ["glass", "white"], "location": "right"}, "garbage_16": {"attributes": ["cylindrical"], "location": "right"}, "doorframe_22": {"attributes": ["rectangular", "white"], "location": "left" }, "oven_24": {"attributes": ["black", "silver"], "location": "right, within arm reach"}... }

example_sit = [
{"user": f "brief situation: sitting on sofa_22, object attributes: {data_sit}",
{"assistant": "'S': 'Sitting on the L-shaped sofa, watching TV far away.', 'O': 'sofa_22, tv_24' 'S': 'Sitting on sofa, chatting with a person on the beanbag to my left.', 'O': 'sofa_22, beanbag_17' 'S': 'Sitting on sofa with a windowsill to the left.', 'O': 'sofa_22, windowsill_4' 'S': 'Sitting on sofa with two plants to the left.', 'O': 'sofa_22, plant_7, plant_8' 'S': 'Sitting on the L-shaped sofa with a wooden table in the front within arm reach.', 'O': 'sofa_22, table_19' 'S': 'Sitting on the L-shaped sofa with two cushions to the left.', 'O': 'sofa_22, cushion_20, cushion_21'" }]

example_interact = [
{"user": f "brief situation: interacting with sink_7, object attributes: {data_interact}",
{"assistant": "'S': 'Washing hands at the sink in the front within arm reach, and a trash can to the left by my feet.', 'O': 'sink_7, trash can_16' 'S': 'Cleaning the sink with a bucket behind me within arm reach.', 'O': 'sink_7, bucket_14' 'S': 'Washing my face at the sink, while the toilet is to my left within arm's reach.', 'O': 'sink_7, toilet_13' 'S': 'Washing hands at the sink with a mirror in the front within arm reach.', 'O': 'sink_7, mirror_9' 'S': 'Washing hands at the sink, with a small shelf to my left.', 'O': 'sink_7, shelf_10'" }]

example_stand = [
{"user": f "brief situation: standing with kitchen counter_2 12 o'clock, object attributes: {data_stand}",
{"assistant": "'S': 'Baking in front of the kitchen counter, with an oven to my right within arm's reach.', 'O': 'kitchen counter_2, oven_24' 'S': 'Cooking in front of the kitchen counter, with the doorframe to my left.', 'O': 'kitchen counter_2, doorframe_22' 'S': 'Cooking in front of the kitchen counter, with a window to my right.', 'O': 'kitchen counter_2, window_13' 'S': 'Cooking in front of the kitchen counter, with a garbage to my right.', 'O': 'kitchen counter_2, garbage_16' 'S': 'Standing in front of the kitchen counter, with two clutters also in the front within arm's reach.', 'O': 'kitchen counter_2, clutter_9, clutter_11'" }]

Figure 9: Prompt for descriptive situation generation.

## A.2 Long-Form Text Generation

Seven coauthors annotated each changed object in 3RScan [17] with situation-agnostic Reason, Description, Rearrangement Instruction, and Warning (Fig. 10). These were combined with object attributes, allocentric and egocentric relationships to generate situation-aware descriptions and instructions.

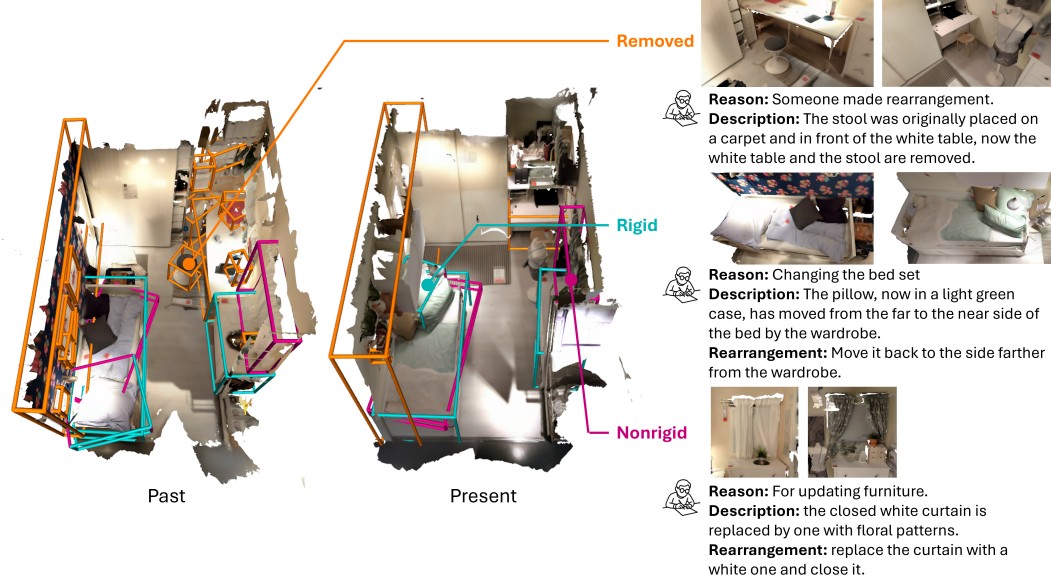

Figure 10: Human annotation for the original changes labeled in 3RScan [17] without situation-awareness. The original changes are categorized into removed, rigid, and non-rigid types.

**system_prompt** = ("You are an AI assistant tasked with generating captions of changes and instructions to rearrange changed objects in a 3D scene, based on the current location and orientation of the observer. This includes the vertical allocentric relationships among the objects, their horizontal locations (specified in degrees and distance) relative to the observer, and their attributes. Objects undergoing changes are classified into four categories: removed, added, rigid, and non-rigid. Always provide a caption ('C') that describes the change, including egocentric details, but exclude any rearrangement instructions ('R') for removed or added objects. To generate caption ('C'), rewrite 'Caption' to include at least one location with distance and clockwise direction: current ('location', 'allocentric') or original ('location_old', 'allocentric_old'), and the distance in 'return'. To generate 'C', don't use direction in 'return'. To generate rearrangement instruction ('R'), rewrite 'Instruction' to guide the user to reach the current 'location' of the changed object for the first step, then do 'return' to return the changed object, *i.e.* at least two steps for 'location' and 'return'. Mention the distance and direction of the movement ('location' and 'return'). And only generate 'C' for objects that have the label 'Caption'. When generating instructions, please always specify the direction and distance of the movement. Please rewrite the numbers (direction and distance) in 'Caption' and 'Instruction' with the provided ones ('location', 'location_old', 'return'), adjust verbs (*e.g.*, push/pull) to reflect the observer's perspective. The output should be formatted as 'O' (object), 'T' (type of change), 'C' (description of change), and 'R' (numbered rearrangement actions, *e.g.*, '1., 2., 3.,...')" )

**data** = {"removed": {"storage_22": { "location_old": "4 o'clock, 0.4m","Caption": "incomplete scan" }}, "rigid": { "table_7": { "attributes": ["wooden", "rectangular", "white"], "location": "10 o'clock, 0.9m", "location_old": "10 o'clock, 1.0m", "allocentric_old": "monitor_8 standing on table_7, picture_23 lying on table_7", "allocentric": "monitor_8 standing on table_7", "Caption": "The table is against the wall, with a computer on top of it, and the window is to the right." }, "chair_6": { "attributes": [ "wide" ], "location": "11 o'clock, 0.8m", "return": "2 o'clock, 0.5m", "location_old": "12 o'clock, 1.0m", "Caption": "The chair was previously by the window, and now it is directly in front of the table.", "Instruction": "Move it one step right to the window" } }, "non_rigid": { "curtain_5": { "location": "11 o'clock, 1.5m", "allocentric": "curtain_5 hanging on wall_3", "Caption": "the change is not obvious" } }, "unchanged": { "rail_33": { "location": "1 o'clock, 1.6m" }...}}
**example** = [
{"user": f "brief situation: standing with chair_34 9 o'clock, object attributes: {data}",
{"assistant": "'O': 'storage_22', 'T': 'removed', 'C': 'The partially scanned storage at your 4 o'clock, 0.6 meter away, may have been removed.' 'O': 'table_7', 'T': 'rigid', 'C': 'The white table with a monitor on it at your 10 o'clock, 1.5 m away hasn't changed its position, but the picture on it has been removed.' 'O': 'chair_6', 'T': 'rigid', 'C': 'The chair, which was at your 11 o'clock, 1.4 meters away by the window, has been moved 0.5 meter to the front of the table.', 'R': 1. Turn to your front-left and take two steps, bypassing the couch half a step away. 2. Pick up the chair in front of the table. 3. Move the chair one step to your right, placing it beneath the window.' 'O': 'curtain_5', 'T': 'nonrigid', 'C': 'The curtain on your 1 o'clock, 1.8 meters away, remains hanging on the wall.'" }]

Figure 11: Prompt for generating situation-aware long-form texts, including change descriptions and rearrangement instructions.

## A.3 Query Generation for Long-Form Text

The queries for change description and rearrangement instruction are generated based on features that distinguish the object from others in the same category, as shown in Fig. 12. These features are filtered and refined by coauthors.

system_prompt = ("You are an AI assistant tasked with generating queries about changes to a specific object. Given the object's name and a set of its features, generate one query per feature. The tense indicates whether the provided information refers to the state before (past) or after (present) the change. Use the tense accordingly when generating queries, especially by referencing the spatial relation of the object (*e.g.*, 'farthest object', 'nearest object', 'others', 'vertical_relationship'). 'others' also represents features of the object. 'num' represents the number of items in the same category within the scene. If 'num' equals 2, use the comparative form for the spatial location; if it is greater than 2, use the superlative form for the spatial location. Don't mention the instance ID of the object. Make the queries as short as possible to include only the necessary information. Please only ask for general changes, and don't ask about the specific change of the object." )

data_pre = {"object": "nightstand_8", "tense": "past", "num":2,
"features": [{"nearest_objs": ["nearest to the curtain"]},
{'vertical_relationships': ["frame standing on nightstand", "lamp supported by nightstand"]}, {"farthest_objs": ["farthest to the wardrobe"]} ]}

data_post = {"object": "desk_5", "tense": "present", "num":3,
"features": [{"nearest_objs": ["nearest to the wardrobe"]},
{'vertical_relationships': ["monitor standing on desk", "plant standing on desk"]} ]}

example_pre= [
"user": {data_pre},
"assistant": {"Query 1": "How has the nightstand that was nearer to the curtain been altered?", "Query 2": "Which updates have been made to the nightstand that had a frame and a lamp on it?", "Query 3": "Could you describe what modifications were applied to the nightstand that stood farther from the wardrobe?", "Query 4": "What changes have been made to the nightstand that stood farther from the wardrobe?", "Query 5": "How has the nightstand that was farther from the wardrobe been altered?", "Query 6": "What changes have been made to the nightstand that stood nearer to the curtain?", "Query 7": "What kind of changes were made to the nightstand set farther from the wardrobe?", "Query 8": "How has the nightstand that was nearer to the curtain been altered?", "Query 9": "Please explain what has been adjusted on the nightstand situated farther from the wardrobe.", "Query 10": "What revisions have taken place regarding the nightstand that was at a distance from the wardrobe?" ]}

example_post= [
"user": {data_post},
"assistant": {"Query 1": "How has the desk that is nearest to the wardrobe been altered?", "Query 2": "What changes have been made to the desk that is closest to the wardrobe?", "Query 3": "Which updates have been made to the desk that has a monitor and a plant on it?", "Query 4": "Could you describe what modifications were applied to the desk with a monitor and a plant on it?", "Query 5": "How has the desk that is positioned nearest to the wardrobe been altered?", "Query 6": "What modifications have been applied to the desk situated nearest to the wardrobe?", "Query 7": "What kind of changes were made to the desk that is closest to the wardrobe?", "Query 8": "How has the desk with a monitor and a plant on it been altered?", "Query 9": "Please explain what has been adjusted on the nightstand situated farther from the wardrobe.", "Query 10": "What revisions have taken place regarding the desk that is closest to the wardrobe?" ]}

Figure 12: Prompt for generating queries for long-form texts, with an example for change description.

## A.4  QA Generation

The QA pairs are generated using object attributes, as well as egocentric and allocentric relationships, following O-CoT [21] as shown in Fig. 13. Each pair may include the object's label, index, and QA type to retrieve the correct answer from the original data. Figure 14 illustrates examples of QA types.

system_prompt = ("You are an AI visual assistant tasked with generating question and answer pairs based on changes observed in a sequence of scene images. The scenes detail the journey along a familiar route, highlighting shifts in object positioning and attributes. Your questions should cover the following areas:
Warning: Query if there is any changed object that obstructs the familiar route to a target object. If an object has the attribute 'Warning' means it becomes an emerged obstacle towards the target object in the list. Only mention one target object in the question. Egocentric Distance Old/ Egocentric Distance ('How far ...'): Calculate the distance from the observer to the current or original location of objects. Prioritize the 'egocentric distance old' if the change exists. Allocentric Displacement ('How far ...'): ask about 'move_distance' of a specific object. Egocentric Direction Old/ Egocentric Direction ('In which direction ...'): Determine the current or original orientation of objects in relation to the observer. Prioritize the 'egocentric direction old' if the change exists. Allocentric Relationship ('Where'): Examine the old or current vertical spatial relationships between objects. Counting: Count objects of a specific type in a direction to the observer (front, left, behind, right). Existence: Note the addition or removal of specific objects. Attribute: Ask about a specific aspect of an object, focusing on its status, color, and material. Questions start with like 'What is the status/ color/ material?'. Affordance: Check for objects serving specific purposes in the observer's immediate vicinity. For each scenario, generate 15 questions and answer pairs addressing these topics to effectively map the changes in the scene. Don't ask anything about the wall, the ceiling, or the floor. Don't answer the direction and distance together. Don't mention numbers in the question. 'Where' is only for an egocentric relationship. Each answer should be a maximum of 5 words. Exclude non-present objects. Don't ask questions that cannot be answered. Don't ask for the direction of the movement. Please don't confuse shape with size. The output is in the format with 'Q' for the question, 'A' for the answer, 'O' for the reference object, and 'Type' for the type of question and answer pairs." )

data = {"rigid": {"chair_39": {"location": "11 o'clock, 0.4m", "move_distance": "1.6m", "location_old": "10 o'clock, 1.7m", "Warning": ["bed_3", "cabinet_6", "bag_24", "bag_38", "blanket_40", "laptop_41", "roll_42"]}, "table_18":{"material": ["wooden"], "color": ["white"], "shape": ["rectangular"], "state": ["messy"], "location": "5'clock, 0.1m", "location_old": "5 o'clock, 0.0m"}}, "unchanged": {"bed_3": {"state": ["messy"], "color":["gray"], "size": ["low", "narrow"], "location": "11 o'clock", 1.1m}, "cabinet_6": {"size": ["big"], "location": "12 o'clock, 1.9m"}, "door_14": { "state": ["closed"], "shape": ["flat"], "color": ["white"],"size": ["tall"],"location": "2 o'clock, 3.9m", "allocentric": "hanging on wall_13"},"clothes_15": {"color": ["beige", "black"], "location": "2 o'clock, 3.8m", "allocentric": "hanging on door_14"}, "chair_19": {"location": "6 o'clock, 0.3m"}, "clothes_dryer_20": {"location": "9 o'clock, 1.9m"}, "window_22": {"material": ["glass"],"shape": ["rectangular"], "state": ["half open/closed"], "location": "9 o'clock, 1.6m", "allocentric": "attached to wall_4, attached to wall_2"},"basket_25": {"size": ["big"],"location": "8 o'clock, 1.4m"},"clothes_dryer_37": {"location": "2 o'clock, 3.1m"}, "blanket_40": {"location": "11 o'clock, 1.8m", "allocentric": " lying on bed_3"}, "laptop_41": {"color": ["gray"],"location": "12 o'clock, 2.4m", "allocentric": "standing on bed_3"}...}

example= [
{"user": {data},
{"assistant": ("'Q': 'How far was the chair, which was between the clothes dryer and the bed, moved?', 'A': '1.6 m', 'O': 'chair_39', 'Type': 'Allocentric Displacement'" "'Q': 'Are there any changed objects on my familiar route to the bed?', 'A': 'A chair', 'O': 'bed_3, chair_39', 'Type': 'Warning'" "'Q': 'What is the status of the white wooden table?', 'A': 'Messy', 'O': 'table_38', 'Type': 'Attribute'" "'Q': 'How far was the clothes dryer to my left relative to me?', 'A': '1.9 m', 'O': 'clothes dryer_20', 'Type': 'Egocentric Distance Old'" "'Q': 'How many chairs are there behind me?', 'A': 'One', 'O': 'chair_19', 'Type': 'Counting'" "'Q': 'Is there something to hang clothes on in this room?', 'A': 'Two clothes dryers', 'O': 'clothes dryer_20, clothes dryer_37', 'Type': 'Affordance'" "'Q': 'Which direction was the changed chair relative to me?', 'A': '10 o'clock', 'O': 'chair_39', 'Type': 'Egocentric Direction Old'" "'Q': 'Is there any sofa in the room?', 'A': 'No', 'O': 'None', 'Type': 'Existence'" "'Q': 'Is there anything to keep warm while sleeping?', 'A': 'A blanket', 'O': 'blanket_40', 'Type': 'Affordance'" "'Q': 'Where are the beige and black clothes?', 'A': 'Hanging on the door', 'O': 'clothes_15, door_14', 'Type': 'Allocentric Relationship'" "'Q': 'Where is the laptop?', 'A': 'Standing on the bed', 'O': 'laptop_41, bed_3', 'Type': 'Allocentric Relationship'" "'Q': 'How far is the basket from me?', 'A': '1.4 m', 'O': 'basket_25', 'Type': 'Egocentric Distance'" "'Q': 'What is the status of the window?', 'A': 'Half open', 'O': 'window_22', 'Type': 'Attribute'" "'Q': 'Which direction is the changed chair relative to me?', 'A': '11 o'clock', 'O': 'chair_39', 'Type': 'Egocentric Direction'" "'Q': 'How far is the chair in front of me?', 'A': '40 cm', 'O': 'chair_39', 'Type': 'Egocentric Distance'") }]

Figure 13: Prompt for QA generation.

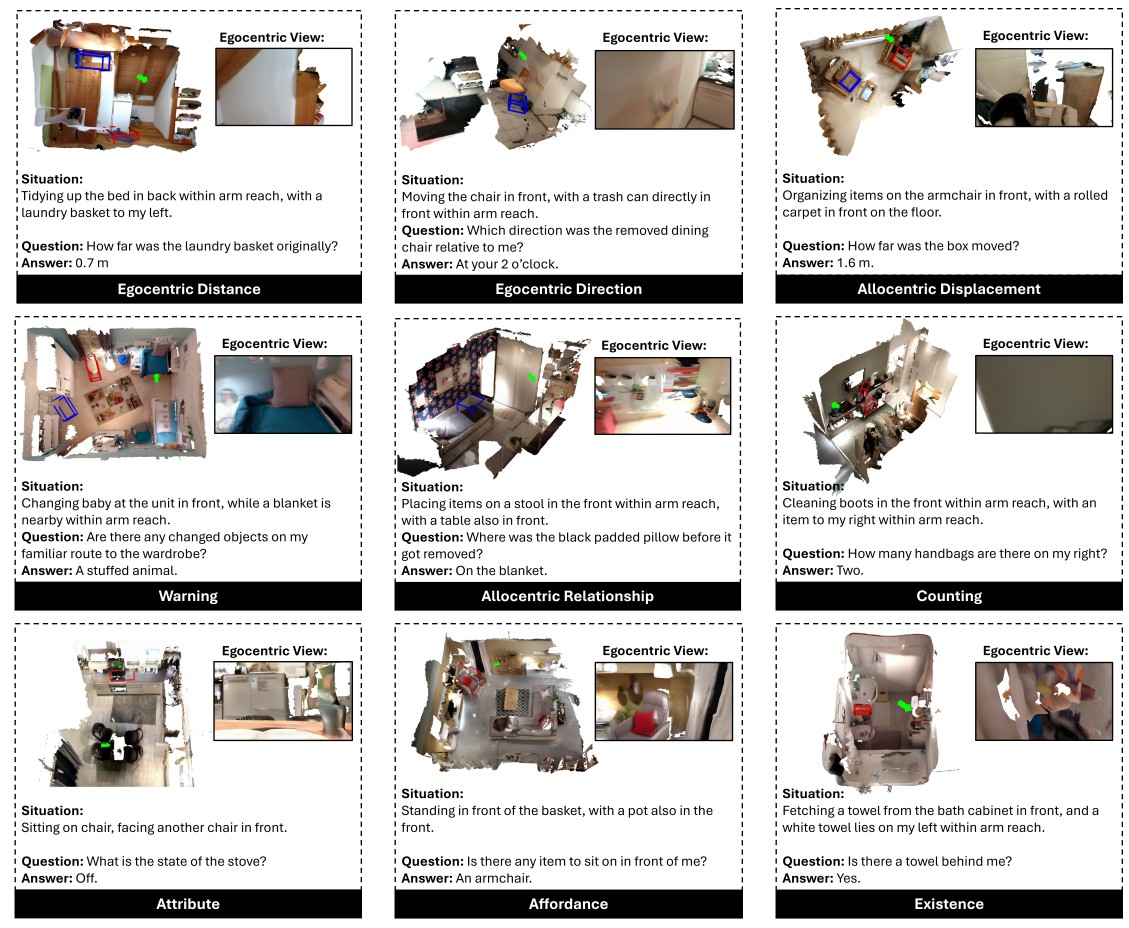

Figure 14: Examples of the nine QA types. Blue indicates the previous position, and red indicates the present position.

# B  Dataset Statistics

Table 6. Statistics of the Situat3DChange dataset on scenes, situations, and changes.

|  | General | | Number of Situations | | | Number of Changes | | |
|---|---|---|---|---|---|---|---|---|
|  | Scan Pairs | Total Objects | Stand | Sit | Interact | Rigid | Removed | Non-Rigid |
| Training | 793 | 22274 | 5981 | 1550 | 3107 | 2543 | 446 | 504 |
| Validation | 110 | 3532 | 882 | 192 | 433 | 390 | 96 | 54 |

Table 7. Distribution of QA types related to scene changes.

| Allo. Dis. | Warning | Ego. Dir. | Allo. Rel. | Attribute | Existence | Ego. Dis. | Affordance | Counting |
|---|---|---|---|---|---|---|---|---|
| 100.00% | 100.00% | 53.41% | 40.03% | 15.88% | 23.79% | 21.66% | 8.91% | 6.91% |

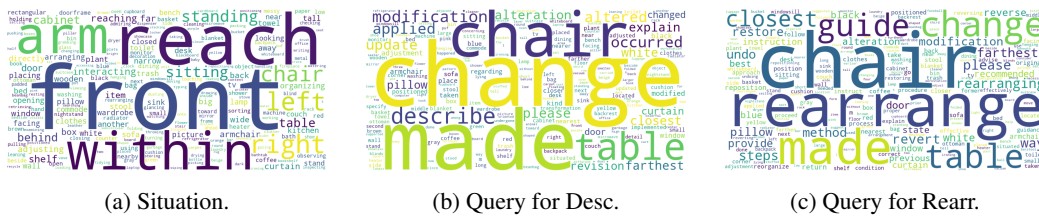

(a) Situation.  (b) Query for Desc.  (c) Query for Rearr.

Figure 15: Word clouds of situations, change description queries, and rearrangement instructions.

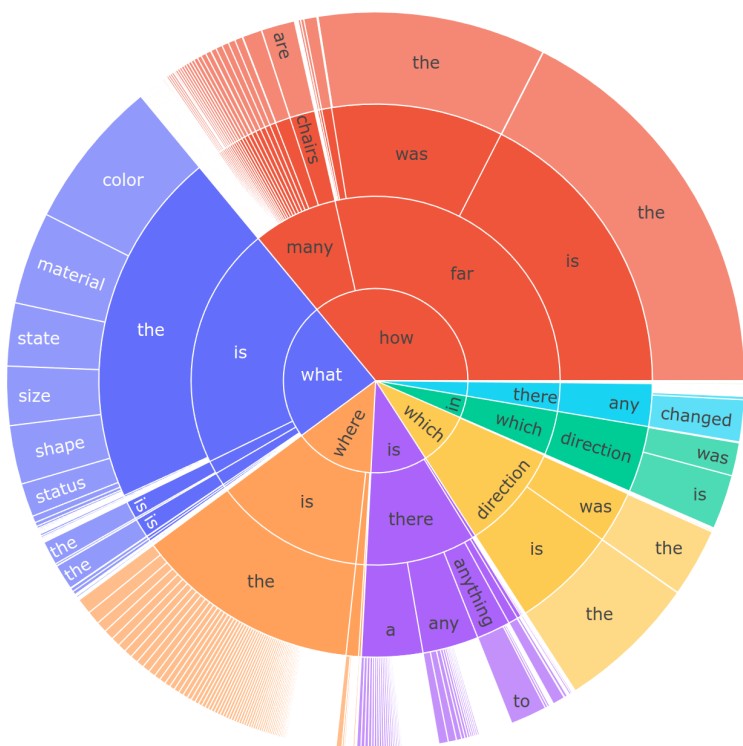

Figure 16: Hierarchical distribution of questions in Situat3DChange.

# C   MLLM Paradigms for Situat3DChange

The paradigms of the baselines are shown in Fig.17, with the tokenizer for system prompts and the decoder for model responses omitted for simplicity. On the right side of Fig.17 (b) is our SCReasoner, which differs from LEO only in the fusion comparison module. This module compares similar tokens from the previous and current scenes, leveraging Mamba's selective token processing [27] and a star operation [28] for fusion. It injects half of the scene tokens into the LLM as LEO does, focusing on the most relevant differences between the scene pair. The hyperparameters for fine-tuning are listed in Tab. 8.

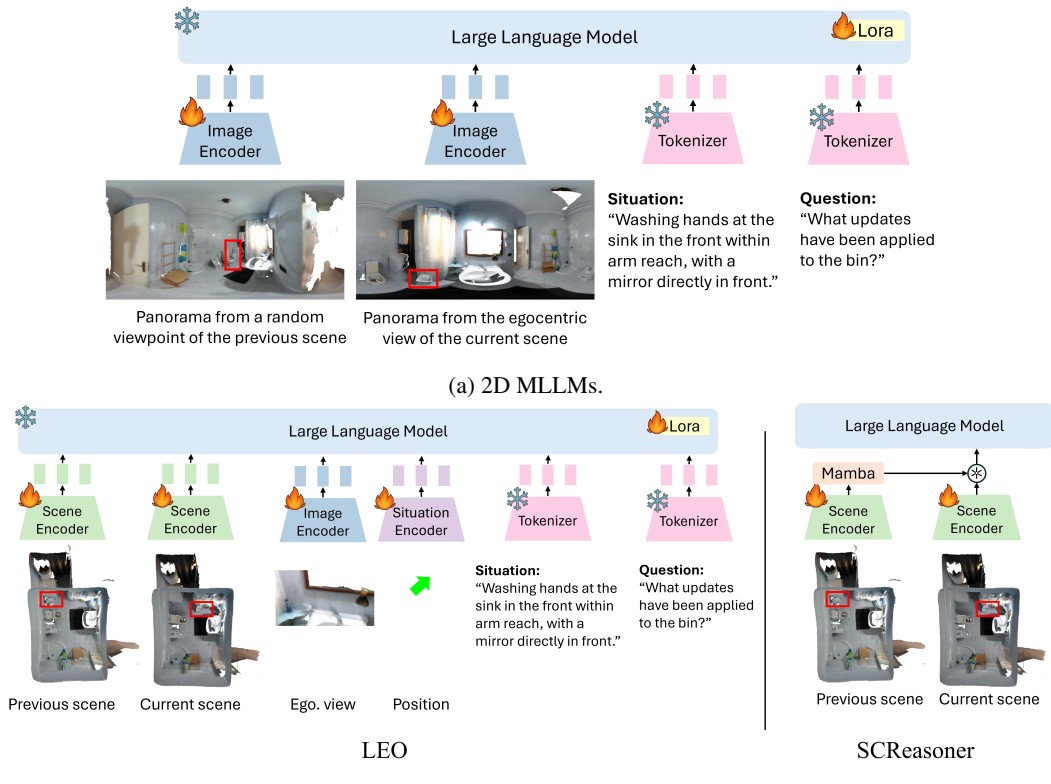

(a) 2D MLLMs.

(b) 3D MLLMs.

Figure 17: Baseline paradigms. Modules with the same color share weights.

Table 8. Hyperparameters for fine-tuned models.

| Hyperparameter | InternVL2 [84] | LEO [21] | SCReasoner |
|---|---|---|---|
| Optimizer | AdamW | AdamW | AdamW |
| Weight decay | 0.05 | 0.05 | 0.05 |
| Betas | $[0.9, 0.999]$ | $[0.9, 0.999]$ | $[0.9, 0.999]$ |
| Learning rate | $4 \times 10^{-5}$ | $3 \times 10^{-5}$ | $3 \times 10^{-5}$ |
| Warmup steps | 2852 | 400 | 400 |
| Number of workers | 4 | 4 | 4 |
| Parallel strategy | DDP | DDP | DDP |
| Type of GPUs | A100 | A100 | A100 |
| Memory | 40 GB | 40 GB | 40 GB |
| Number of GPUs | 4 | 4 | 4 |
| Batch size per GPU | 2 | 2 | 2 |
| Training precision | bfloat16 | bfloat16 | bfloat16 |
| Epochs | 5 | 5 | 5 |

# D Evaluation Details

## D.1 Prompts for GPT Scoring

We craft fine-grained prompts to evaluate open-ended responses from MLLMs: general QA in Fig.18, direction-related QA in Fig.19, and long-form text in Fig. 11. For evaluation, we use GPT-4o-mini [92] (timestamp 2024-07-18), which has been shown to outperform GPT-3.5 in both accuracy and cost-efficiency.

---

Score open-ended answers from 1 to 5 based on accuracy to the ground truth.
Score 2-4: Reflect partial correctness or minor errors.
Criteria:
Affordance: Question: Is there any furniture to rest feet on nearby? Ground Truth: Yes. Example Response: Yes, there is an ottoman nearby. Score: 5 (Correct match). Attribute: Question: What is the color of the ottoman? Ground Truth: Blue, red, brown. Example Response: The ottoman is brown. Score: 3 (Partial match). Existence: Question: Is there a chair on my left? Ground Truth: Yes. Example Response: Yes, there is a chair on the left. Score: 5 (Correct match). Counting: Question: How many tables are in the room? Ground Truth: Three Examples. Response: Two. Score: 1 (Significant discrepancy). Warning: Question: Are there any changed objects on my familiar route to the door? Ground Truth: Yes, a chair. Example Response: Yes, there is a table on the way to the door. Score: 2 (Major incorrect). Allocentric Relationship: Question: Where is the kettle? Ground Truth: On the kitchen cabinet. Example Response: The kettle is on the kitchen counter. Score: 4 (Approximate match).
Output only the score.

---

Figure 18: Prompt for LLM-assisted scoring of general QA.

---

Score open-ended answers from 1 to 5 based on accuracy to the ground truth.
Score 2-4: Reflect partial correctness or minor errors.
Mapping of proximity direction and clock face: front (from 11 to 1 o'clock), left (from 8 to 10 o'clock), right (from 2 to 4 o'clock), back (from 5 to 7 o'clock).
Criteria:
Score 5: If the difference is less than or equal to 1 o'clock on the clock face, *e.g.*, GT: '11 o'clock', Response: '10 o'clock'. Score 4: If the response is in the correct proximity direction, *e.g.*, GT: '6 o'clock'(back), Response: 'Back'. Score 3: If the response is adjacent to the correct direction, *e.g.*, GT: '11 o'clock'(front left), Response: 'Left'. Score 2: If the response has a significant directional error but is not completely opposite, *e.g.*, GT: '3 o'clock'(right), Response: 'Back'. Score 1: If the response is in the opposite proximity direction to the ground truth, *e.g.*, GT: '9 o'clock'(left), Response: '4 o'clock'(right).
Output only the score.

---

Figure 19: Prompt for LLM-assisted scoring of egocentric direction QA.

---

You are an intelligent evaluator tasked with assessing the correctness and semantic similarity of model-generated answers to question-answering pairs. Your goal is to compare the predicted answer with the reference (correct) answer and assign a score based on how well they align in meaning. Use the following scoring rubric:
Score 5: Completely correct or semantically equivalent.
Score 4: Key information is correct, with minor inaccuracies or omissions.
Score 3: Some relevant information, but lacks sufficient correctness or completeness.
Score 2: Mostly incorrect, but shows some relevance to the question.
Score 1: Completely incorrect or nonsensical.
Your response must be a single integer from 1 to 5, with no additional text or explanation.

---

Figure 20: Prompt used for LLM-assisted scoring of long-form texts, including change descriptions and rearrangement instructions.

### D.2 Alignment between Human and GPT Evaluation

To validate the validity of the GPT-based evaluation results, we recruited and acknowledged four human evaluators who are not involved in this project. We selected SCReasoner, LEO, and one-shot InternVL for human evaluation. SCReasoner and LEO are compared to demonstrate our improvements, while one-shot InternVL represents an open-ended, training-free LLM. For each model, we randomly selected 40 samples for each QA type (excluding those related to distance) and each long-form task, resulting in the same 360 samples per model. As shown in the table, SCReasoner consistently outperforms LEO on the 360 sampled instances, with the performance gap further amplified when evaluated by human scores compared to GPT scores.

Table 9. Human and GPT evaluation results.

| Model | Description | | Rearrangement | | QA | |
|---|---|---|---|---|---|---|
| | GPT | Human | GPT | Human | GPT | Human |
| InternVL | 4.0 | 6.5 | 3.0 | 7.5 | 33.1 | 36.4 |
| LEO | 11.5 | 14.5 | 22.5 | 19.5 | 43.3 | 45.6 |
| SCReasoner | 14.0 | 20.5 | 26.0 | 31.5 | 48.3 | 50.9 |

Following OpenEQA [71], we computed the Spearman correlation between human scores and GPT-generated scores. The GPT scores show a strong correlation with human evaluation ($\rho > 0.6$), indicating that GPT-based evaluation aligns well with human judgment.

Table 10. Spearman correlation between GPT and human evaluations.

| | Description | Rearrangement | QA | Average |
|---|---|---|---|---|
| Spearman Corr. | 0.75 | 0.70 | 0.94 | 0.80 |

# E  Ablation Studies

## E.1  Consistent Improvements with Run-to-Run Validation

In the main text, we followed LEO's setting by fixing the random seed. To better analyze the errors and demonstrate the performance gain of SCReasoner over LEO, we conducted three additional runs with different random seeds, resulting in four runs in total. The mean scores and standard deviations for each task are reported in Tab. 11, showing that SCReasoner consistently outperforms LEO.

Table 11. Results of LEO and SCReasoner (mean ± std over 5 seeds).

|  | QA | Description | Rearrangement |
|---|---|---|---|
| LEO | $52.760 \pm 0.511$ | $12.733 \pm 0.117$ | $30.165 \pm 0.410$ |
| SCReasoner | $53.420 \pm 0.232$ | $13.629 \pm 0.287$ | $30.753 \pm 0.216$ |

## E.2  Panorama vs. Multiview Input for 2D MLLMs

Since 2D MLLMs are rarely trained on panoramas and are only weakly exposed to multi-image understanding, we stitched the four surrounding views with the front view placed at the center to form a single surround-view image as input to the one-shot InternVL model, in order to assess the plausibility of using panoramas. As shown in Table 12, the performance difference between panorama and stitched multi-view inputs is negligible.

Table 12. Comparison of panorama and multi-view inputs for 2D MLLM performance.

| Input | QA | Description | Rearrangement |
|---|---|---|---|
| panorama | 35.7 | 3.8 | 3.7 |
| multi-view | 35.5 | 3.7 | 3.5 |

## E.3  Understanding the Underperformance of 2D MLLMs

The short QA results in Tab. 3 can be regarded as a decomposition of the comprehensive understanding of dynamic situations and environments. We observe that 2D MLLM performs better than 3D MLLMs in terms of egocentric information with a similar model size, while underperforming in allocentric reasoning and object property identification. This indicates that although panoramic views can effectively reconstruct visible areas and convey orientation information [85], they still suffer from occlusion issues and limited field of view constraints.

To further analyze this limitation, we report results in Tab. 13 for long-form tasks using two 2D MLLMs with one-shot CoT prompting, compared against one-shot results without CoT. We prompt the models to separately analyze the two panoramic views and then compare them based on the current situation, in order to better capture allocentric information. This strategy yields notable performance gains for both models, except that Janus remains on par in the rearrangement task, underscoring the importance of allocentric reasoning. Nevertheless, their performance still lags behind parameter-efficient fine-tuned counterparts.

Table 13. Effect of chain-of-thought (CoT) prompting on description and rearrangement tasks.

| Model | Description (wo CoT) | Description (with CoT) | Rearrangement (wo CoT) | Rearrangement (with CoT) |
|---|---|---|---|---|
| InternVL | 3.8 | 5.3 | 3.7 | 4.9 |
| Janus | 2.7 | 3.6 | 4.7 | 4.3 |

