# Situat3DChange: Situated 3D Change Understanding Dataset for Multimodal Large Language Model (Supplementary Materials)

**Ruiping Liu**[1]     **Junwei Zheng**[1]     **Yufan Chen**[1]     **Zirui Wang**[1]     **Kunyu Peng**[1]
**Kailun Yang**[2]     **Jiaming Zhang**[1,3,*]     **Marc Pollefeys**[3]     **Rainer Stiefelhagen**[1]
[1] Karlsruhe Institute of Technology (KIT)     [2] Hunan University     [3] ETH Zurich

## A   Data Generation

The data generation process includes situation sampling, long-form text generation, query generation for the long-form text, and QA generation. It is based on human observations of changes, object attributes, and allocentric object relationships in 3DSSG [9], as well as egocentric relationships between the human and the objects.

### A.1   Situation Sampling

We follow the situation categories of MSQA [4], namely sitting, interacting, and standing, but with more detailed geometric analysis:

**Sitting.** The 28 seat categories in 3RScan [8] are grouped into four types: 3 large seats with backrests (*e.g.*, sofa), 16 small seats with backrests (*e.g.*, armchair), 1 large seat without a backrest (bed), and 8 small seats without backrests (*e.g.*, beanbag). Seatable and backrest areas are classified by surface normals, or by nearby walls within 0.5 m if no backrest exists. For small seats, the seating point is the bounding box center, oriented away from the backrest. For large seats, we select a point with a backrest behind and open space (0.5–1 m) in front. If no backrest or wall is present, the seat center is used, facing the room center.

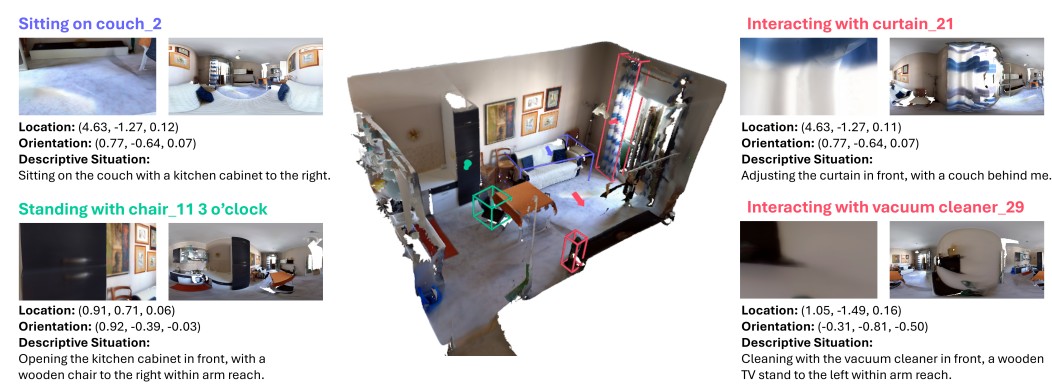

**Sitting on couch_2**
**Location:** (4.63, -1.27, 0.12)
**Orientation:** (0.77, -0.64, 0.07)
**Descriptive Situation:**
Sitting on the couch with a kitchen cabinet to the right.

**Standing with chair_11 3 o'clock**
**Location:** (0.91, 0.71, 0.06)
**Orientation:** (0.92, -0.39, -0.03)
**Descriptive Situation:**
Opening the kitchen cabinet in front, with a wooden chair to the right within arm reach.

**Interacting with curtain_21**
**Location:** (4.63, -1.27, 0.11)
**Orientation:** (0.77, -0.64, 0.07)
**Descriptive Situation:**
Adjusting the curtain in front, with a couch behind me.

**Interacting with vacuum cleaner_29**
**Location:** (1.05, -1.49, 0.16)
**Orientation:** (-0.31, -0.81, -0.50)
**Descriptive Situation:**
Cleaning with the vacuum cleaner in front, a wooden TV stand to the left within arm reach.