# OpenReview forum: "Situat3DChange: Situated 3D Change Understanding Dataset for Multimodal Large Language Model"
_NeurIPS.cc/2025/Datasets_and_Benchmarks_Track — NeurIPS 2025 Datasets and Benchmarks Track poster_

### Official Review · Reviewer_KWpT · 2025-06-30

**Rating:** 6
**Confidence:** 4

**Summary:**

This paper introduces a large-scale dataset and benchmark tailored for situation-aware 3D scene change understanding. It includes over 174k samples derived from real-world scene scan pairs, annotated for three tasks: QA (121k), change description (36k), and rearrangement instruction generation (17k).

In addition, the authors propose a 3D multimodal language model that performs token-efficient point cloud comparison. Experiments show strong improvements over prior MLLMs, and demonstrate promising scaling and cross-domain transfer effects.

**Dataset Code Accessibility:**

Yes

**Dataset Code Comments:**

Please refer to https://huggingface.co/datasets/lrp123/Situat3DChange

**Ethical Considerations:**

No, there are no or only very minor ethics concerns

**Final Justification:**

The author's rebuttal has solved my questions, and I'll keep the original score for accepting the paper.

**Limitations Weaknesses:**

1. Instructions use imprecise descriptors like "steps" and "left/right", making the task more ambiguous and harder to evaluate consistently.

**Strengths Contributions:**

1. The paper highlights an underexplored yet critical aspect of 3D scene understanding: joint modeling of scene dynamics and situational awareness.
2. The dataset is rich in annotations, covering 3D change detection, perception and action.
3. Annotation by experts with experience in assisting visually impaired users makes the dataset aligned with real-world cognition and accessibility concerns.
4. The proposed SCReasoner introduces an efficient architecture for comparing similar point cloud pairs.

---

> ### Author Rebuttal · Authors · 2025-07-30
>
> Dear Reviewer KWpT,
>
> thank you for recognizing the uniqueness of our dataset in jointly modeling dynamic environments and situational awareness, as well as the human effort involved in annotating the three tasks from perception to action. Below, we address your thoughtful concern.
>
> **[W1] Vague instructions reduce consistency**
> >Instructions use imprecise descriptors like "steps" and "left/right", making the task more ambiguous and harder to evaluate consistently.
>
> The dataset is indeed more challenging. The use of imprecise descriptors is informed by human-centered research and aims to facilitate shared situational awareness and mental mapping between humans and embodied agents, thereby supporting human–AI collaboration. Below are several additional relevant studies that we will cite to support this design choice:
>
> [1] *Assessment of Indoor Route-Finding Technology for People with Visual Impairment* (J Vis Impair Blind, 2010)
>
> [2] *Virtual Navigation for Blind People: Transferring Route Knowledge to the Real World* (IJHCS, 2020)
>
> [3] *What Do Blind and Low-Vision People Really Want from Assistive Smart Devices? Comparison of the Literature with a Focus Study* (ASSETS, 2023)
>
> [4] *Wayfinder Design Principles for Vision-Impaired People* (Website)

---

> > ### Comment · Reviewer_KWpT · 2025-08-09
> >
> > The author's rebuttal has solved my questions, and I'll keep the original score for accepting the paper.

---

### Official Review · Reviewer_fa1y · 2025-07-03

**Rating:** 4
**Confidence:** 4

**Summary:**

The paper introduces an interesting benchmark that couples scene change with situational (egocentric + allocentric) context, it offers a more realistic real-world setting compared to traditional static scene understanding tasks. This is especially relevant, as indoor scenes are often dynamic, with objects being rearranged over time. The paper also introduces a large-scale dataset containing 121k QA pairs, 36k descriptions of object changes within scenes, and 17k rearrangement instructions to support the tasks. Additionally, the paper proposes an efficient baseline method designed to reduce token redundancy while tackling the task. Experimental results highlight the challenges faced by traditional VLMs and 3D VLMs, which often struggle with the proposed task due to its demands for deeper 3D spatial understanding and reasoning.

**Dataset Code Accessibility:**

Yes

**Dataset Code Comments:**

The dataset is publicly available on Hugging Face, and the code is released on GitHub.

**Ethical Comments:**

The proposed dataset and benchmark are built upon the existing public 3D scene dataset 3RScan. I do not identify any significant ethical concerns

**Ethical Considerations:**

No, there are no or only very minor ethics concerns

**Final Justification:**

This paper presents an interesting perspective on 3D scene understanding, focusing on interpreting scene changes from both egocentric and allocentric viewpoints, the setting is particularly relevant for robotics applications. The original manuscript lacked two important statistical tables. However, the authors have provided them in the rebuttal and committed to including them in the final version. I am inclined to maintain my original score (4: Borderline Accept).

**Limitations Weaknesses:**

1. The paper lacks discussion of several recent and relevant 3D VQA benchmarks, such as Thinking in Space, 3D-GRAND, M3DBench, and MMScan. Including comparisons or discussions with these works would provide better context for the paper’s contributions.
2. The choice of InternVL2-7B for fine-tuning is unclear. Given that it underperforms other 2D VLMs in zero- and one-shot settings, its inclusion raises fairness concerns. It remains questionable whether the proposed SCReasoner would still outperform a stronger fine-tuned 2D VLM.
3. It is unclear how many QA samples in the dataset involve object changes. Some questions, such as "What is the color of the armchair?", seem unrelated to object changes. Providing statistics on the proportion of questions involving object changes is essential, especially since modeling object changes is claimed as the main contribution.
4. Minor issue: the paper does not clearly explain how the panorama images were obtained, whether they are rendered from reconstructed meshes or extracted from raw video frames. This detail is important for reproducibility and understanding the data quality for evaluation.

**Strengths Contributions:**

1. The benchmark introduces a novel setting and addresses an important research problem, evaluating how well a model can understand spatial changes of objects in 3D scenes (with decimeter-level accuracy), how well a model can measure the safety and rationality of an environment arrangement and how closely a model’s reasoning aligns with human cognition. This sets up a significant research question in the field of 3D scene understanding and reasoning.
2. The proposed dataset is large-scale and includes extensive human annotations. Ablation studies (Table 4, 5) demonstrate that using this dataset does not compromise a model’s general ability to perform traditional QA and captioning tasks.
3. The proposed baseline method is both intuitive and effective. It also includes comprehensive comparisons with both 2D and 3D VLMs, serving as a strong reference point for future research.
4. The paper provides sufficient details regarding the dataset and benchmark construction, supporting reproducibility and further study.

---

> ### Author Rebuttal · Authors · 2025-07-30
>
> Dear Reviewer fa1y,
>
> thank you for recognizing the significance of our dataset in bridging dynamic scene changes with situational context, as well as the token-efficient MLLM we developed to address the task. Below, we respond to your concerns.
>
> **[W1] Missing related work discussion**
> >The paper lacks discussion of several recent and relevant 3D VQA benchmarks, such as Thinking in Space, 3D-GRAND, M3DBench, and MMScan. Including comparisons or discussions with these works would provide better context for the paper’s contributions.
>
> We’re grateful for your recommendation. While these recent works are related to situated 3D reasoning, they differ significantly from our dataset. We will update Table 1 as below for clear comparison, and discuss the datasets in the first section of related work: "Thinking in Space introduces a dataset designed to facilitate spatial understanding from short egocentric videos. 3D-GRAND is a large-scale, densely grounded 3D dataset aimed at mitigating hallucinations. M3DBench and MMScan address situated scene understanding from global to local perspectives, but are limited in terms of embodied agent viewpoints and static scenarios[...] In contrast to previous situational understanding datasets focused on static scenes, we introduce Situat3DChange, a dataset for 3D scene understanding in dynamic scenarios and situations."
>
> | Dataset             | Change | Perspective | Data Type | Scenes      | Annotation           | QA Types | QA Pairs | Description | Rearrangement |
> |---------------------|--------|-------------|-----------|-------------|----------------------|----------|----------|-------------|---------------|
> | Dy2Change           | ✓      | allo.       | 3D sim    | 38K pairs   | Template             | -        | -        | 661K        | -             |
> | EmbSCU              | ✓      | allo.       | 2D sim    | 20.5K pairs | Template             | -        | -        | 21K         | 21K           |
> | PanoSCU             | ✓      | allo.       | 3D sim    | 2650 pairs  | Template             | -        | -        | 9K          | 9K            |
> | ReferIt3D           | ✗      | allo.       | 3D scan   | 707 scenes  | Human                | -        | -        | 41.5K       | -             |
> | ScanRefer           | ✗      | allo.       | 3D scan   | 800 scenes  | Human                | -        | -        | 51.6K       | -             |
> | 3D-GRAND            | ✗      | allo.       | 3D scan   | 40K scans   | Template+GPT         | 6        | 5.5M     | 118K        | -             |
> | MMScan              | ✗      | allo.       | 3D scan   | 5.2K scans  | Template+GPT+Human   | 5        | 1.76M    | 4.06M       | -             |
> | OpenEQA             | ✗      | allo.       | 3D scan   | 180 scans   | Human                    | 7        | 1.6K     | -           | -             |
> | SQA3D               | ✗      | ego.        | 3D scan   | 650 scans   | Human                | 6        | 33.4K    | -           | -             |
> | MSQA                | ✗      | ego.        | 3D scan   | 1734 scans  | GPT                  | 9        | 251K     | -           | -             |
> | VSI-Bench           | ✗      | ego.+allo.  | 2D video  | 288 videos  | Human                | 8        | 5K       | -           | -             |
> | RoboSpatial         | ✗      | ego.+allo.        | 3D scan   | 5223 scans  | Human                | 9        | 3M       | -           | -             |
> | Situat3DChange (Ours)| ✓     | ego.+allo.  | 3D scan   | 903 pairs   | Human+GPT            | 9        | 121K     | 36K         | 17K           |
>
> **[W2] Unclear InternVL2 choice**
> >The choice of InternVL2-7B for fine-tuning is unclear. Given that it underperforms other 2D VLMs in zero- and one-shot settings, its inclusion raises fairness concerns. It remains questionable whether the proposed SCReasoner would still outperform a stronger fine-tuned 2D VLM.
>
> Thanks for your concern. In fact, InternVL does not underperform all other models, but rather performs comparably. In the one-shot setting, the best-performing open-source 2D MLLMs vary across the three tasks: InternVL achieves the highest performance on change description, Qwen on rearrangement instruction, and DeepSeek on QA. Therefore, for fine-tuning, we selected InternVL as it is the state-of-the-art open-source MLLM on the widely used MMMU benchmark (CVPR'24) (see Lines 246–248). We adopted a unified fine-tuning strategy to maintain consistency across tasks and avoid fragmenting the training pipeline across multiple models. InternVL’s strong general performance makes it a suitable and stable backbone for all three tasks.
>
>
> **[W3] No object change stats**
> >It is unclear how many QA samples in the dataset involve object changes. Some questions, such as "What is the color of the armchair?", seem unrelated to object changes. Providing statistics on the proportion of questions involving object changes is essential, especially since modeling object changes is claimed as the main contribution.
>
> Thank you for your suggestion. We wrote a script to analyze keywords related to change and verb tense. The statistics on change-related questions are provided in the table below, and the indices of the corresponding QA samples will be released.
>
> | Allocentric Displacement | Warning | Egocentric Direction | Allocentric Relationship | Attribute | Existence | Egocentric Distance | Affordance | Counting |
> |:-----------------------------:|:-----------:|:-------------------------:|:----------------------------:|:-------------:|:-------------:|:------------------------:|:--------------:|:------------:|
> | 100.00%                       | 100.00%     | 53.41%                    | 40.03%                       | 15.88%        | 23.79%        | 21.66%                   | 8.91%          | 6.91%        |
>
> Note that our dataset is designed not only for dynamic scenarios, but also for dynamic situations, supporting holistic scene understanding. While not all QA samples are directly related to change, all of them contribute to comprehensive scene understanding.
>
> **[W4] Panorama generation unclear**
> >Minor issue: the paper does not clearly explain how the panorama images were obtained, whether they are rendered from reconstructed meshes or extracted from raw video frames. This detail is important for reproducibility and understanding the data quality for evaluation.
>
> The panoramas are generated by stitching skybox images rendered from the scene, following Matterport3D (3DV'17). The tool for skybox capture and instructions for generating panoramas will be made available in the GitHub repository, while the resulting panoramas are already hosted in the Hugging Face repository.

---

> > ### Comment · Reviewer_fa1y · 2025-08-04
> >
> > Thanks for the authors’ rebuttal. I encourage the incorporation of the above statistics into the manuscript. I have no further questions.

---

> > > ### Author Response · Authors · 2025-08-04
> > > **Thank you**
> > >
> > > Thank you for your thoughtful feedback and support. We're glad that we have addressed all your concerns. We will incorporate the suggested statistics into the revised manuscript as recommended. We appreciate your time and consideration.

---

### Official Review · Reviewer_3iRY · 2025-07-03

**Rating:** 4
**Confidence:** 4

**Summary:**

The paper introduces Situat3DChange, a large-scale dataset designed to address the gap in understanding dynamic 3D environments from a situated, or egocentric, perspective. Built upon 11,000 human observations of environmental changes, the dataset provides 121,000 question-answer pairs, 36,000 change descriptions, and 17,000 rearrangement instructions to support perception and action tasks. To effectively utilize this data, the authors also propose SCReasoner, a novel and efficient Multimodal Large Language Model (MLLM) architecture that compares pairs of similar point clouds by focusing on their differences with minimal parameter overhead. Through comprehensive experiments, the paper demonstrates the dataset's value for training more capable models and shows that SCReasoner improves performance on tasks requiring an understanding of spatial changes, with further analysis confirming positive scaling effects and cross-domain transferability.

**Additional Feedback:**

No

**Dataset Code Accessibility:**

Yes

**Dataset Code Comments:**

I can access the code and data easily.

**Ethical Comments:**

I didn't identify siginificant ethical concerns in the paper.

**Ethical Considerations:**

No, there are no or only very minor ethics concerns

**Final Justification:**

The author's rebuttal has solved my questions, and I'll keep the original score for accepting the paper.

**Limitations Weaknesses:**

1. While SCReasoner outperforms the LEO baseline, the margin on certain tasks is quite small. For example, in the rearrangement instruction task, the GPT-score improves from 30.1 for LEO to 30.7 for the best SCReasoner variant. Without statistical significance testing (e.g., error bars from multiple runs), it is difficult to be certain that this 0.6% improvement is meaningful rather than a result of run-to-run variance. This is particularly relevant for a key contribution of the paper.
2. The paper claims SCReasoner is token-efficient because it requires "no additional tokens required for the language decoder". This is true for the decoder's input, but the overall architecture still processes two full point clouds through an encoder, which is computationally more demanding than a single-scene model. The primary benefit seems to be preventing the  decoder from being overwhelmed by redundant information, which improves its ability to focus on changes. The claim could be refined to more precisely state that the efficiency gain is in the decoding step and in improving the relevance of the tokens presented to the LLM, rather than overall computational efficiency.
3. The primary metric for the QA and long-form tasks is a GPT-based correctness score. It is known that LLM evaluators can exhibit biases, such as a preference for longer or more verbose answers (which the authors acknowledge) or favoring outputs that share a similar style to their own training data. This could create a situation where the model being evaluated is inadvertently optimized to please the style of the evaluator model. Maybe human and GPT evaluation alignment is required.
4. The model architecture difference between SCReaonser and LEO should be further explained in the paper in detail. Besides, SCReasoner has four different variants: Linear+, Linear*, Mamba+, and Mamba*, which should be introduced in a clearer manner.
5. For 2D MLLMs, can we directly use the multi-view images instead of the panoramas for scene understanding? I think the 2D MLLMs are rarely trained on the panoramas.

**Strengths Contributions:**

1. The paper successfully identifies and tackles a crucial, underexplored area at the intersection of 3D vision, language, and robotics. The focus on integrating dynamic scene understanding with situated, human-centric awareness is highly relevant for the future of embodied AI and human-AI collaboration.
2. The methodology for creating the dataset is a major strength. By starting with 11,000 detailed annotations from seven co-authors experienced in assisting visually impaired people, the authors ground their dataset in genuine human perception.
3. The three proposed tasks (QA, Description, Rearrangement) are well-structured and cover a spectrum of understanding from perception to action. The mix of concise answers (QA) and long-form text (descriptions, instructions) creates a rich and challenging benchmark.

---

> ### Author Rebuttal · Authors · 2025-07-30
>
> Dear Reviewer 3iRY,
>
> thank you for recognizing our dataset's integration of dynamic scene understanding with situated, human-centric awareness, and for noting how we structured the tasks from perception to action. Below, we address your concerns.
>
> **[W1] Marginal performance improvement**
> >While SCReasoner outperforms the LEO baseline, the margin on certain tasks is quite small. For example, in the rearrangement instruction task, the GPT-score improves from 30.1 for LEO to 30.7 for the best SCReasoner variant. Without statistical significance testing (e.g., error bars from multiple runs), it is difficult to be certain that this 0.6% improvement is meaningful rather than a result of run-to-run variance. This is particularly relevant for a key contribution of the paper.
>
> Thank you for your suggestions and we will add the run-to-run analysis in the camera-ready version. In our original submission, we followed LEO’s setting by fixing the random seed. To better analyze the errors and demonstrate the performance gain of SCReasoner over LEO, we conducted three additional runs with different random seeds, resulting in four runs in total. The mean scores and standard deviations for each task are reported below, showing that SCReasoner consistently outperforms LEO.
> |              | QA             | Description     | Rearrangement    |
> |--------------|--------------------|----------------------|-----------------------|
> | LEO      | 52.760 ± 0.511     | 12.733 ± 0.117       | 30.165 ± 0.410        |
> | SCReasoner | 53.420 ± 0.232     | 13.629 ± 0.287       | 30.753 ± 0.216        |
>
>
> **[W2] Refine token-efficiency claim**
> >The paper claims SCReasoner is token-efficient because it requires "no additional tokens required for the language decoder". This is true for the decoder's input, but the overall architecture still processes two full point clouds through an encoder, which is computationally more demanding than a single-scene model. The primary benefit seems to be preventing the decoder from being overwhelmed by redundant information, which improves its ability to focus on changes. The claim could be refined to more precisely state that the efficiency gain is in the decoding step and in improving the relevance of the tokens presented to the LLM, rather than overall computational efficiency.
>
> Thank you for the valuable input. We will refine the claim as "To adapt to the Situat3DChange tasks, we introduce SCReasoner, an MLLM architecture that compares scene pairs with a selective fusion projector, thereby reducing the number of tokens passed to the language decoder and preventing it from being overwhelmed by redundant scene information."
>
> **[W3] GPT evaluation bias**
> >The primary metric for the QA and long-form tasks is a GPT-based correctness score. It is known that LLM evaluators can exhibit biases, such as a preference for longer or more verbose answers (which the authors acknowledge) or favoring outputs that share a similar style to their own training data. This could create a situation where the model being evaluated is inadvertently optimized to please the style of the evaluator model. Maybe human and GPT evaluation alignment is required.
>
> We value this suggestion and will revise accordingly. We recruited four human evaluators who are not involved in this project and will acknowledge them in the paper. We selected SCReasoner, LEO, and one-shot InternVL for human evaluation. SCReasoner and LEO are compared to demonstrate our improvements, while one-shot InternVL represents an open-ended, training-free LLM. For each model, we randomly selected 40 samples for each QA type (excluding those related to distance) and each long-form task, resulting in the same 360 samples per model. As shown in the table, SCReasoner consistently outperforms LEO on the 360 sampled instances, with the performance gap further amplified when evaluated by human scores compared to GPT scores.
> | Model     | Description GPT | Description Human | Rearrangement GPT | Rearrangement Human | QA GPT | QA Human |
> |:-------------:|:-------------------:|:----------------------:|:----------------------:|:------------------------:|:----------:|:------------:|
> | InternVL      | 4.0                 | 6.5                    | 3.0                    | 7.5                      | 33.1       | 36.4         |
> | LEO           | 11.5                | 14.5                   | 22.5                   | 19.5                     | 43.3       | 45.6         |
> | SCReasoner| 14.0            | 20.5               | 26.0               | 31.5                 | 48.3   | 50.9     |
>
>
>
> Following OpenEQA (CVPR'24), we computed the Spearman correlation between human scores and GPT-generated scores. The GPT scores show a strong correlation with human evaluation (ρ > 0.6), indicating that GPT-based evaluation aligns well with human judgment.
>
> | | Description | Rearrangement | QA  | Average |
> |:----------------:|:---------------:|:-----------------:|:-------:|:-------:|
> | Spearman Corr. |      0.75       |       0.70        |  0.94   |  0.80   |
>
> **[W4] Unclear model variants**
> >The model architecture difference between SCReaonser and LEO should be further explained in the paper in detail. Besides, SCReasoner has four different variants: Linear+, Linear*, Mamba+, and Mamba*, which should be introduced in a clearer manner.
>
> Thanks for your suggestions, we will clarify the difference between SCReasoner and LEO in Section 4, 3D MLLM: "There is no existing MLLM that takes two point clouds as input. We implemented LEO ourselves by simply prepending tokens from the two point clouds at the input, same as how tokens from other modalities are handled. In contrast, SCReasoner employs a selective comparison projector before decoder to improve token efficiency. It comprises a projection module to better focus on scene changes, and a fusion module to eliminate redundancy. For the projection module, we considered a linear layer for simple channel-wise projection and Mamba for selective projection. For parameter-free fusion, we experimented with element-wise addition (+) and multiplication (\*)."
>
>
> **[W5] Panorama vs. multi-view**
> > For 2D MLLMs, can we directly use the multi-view images instead of the panoramas for scene understanding? I think the 2D MLLMs are rarely trained on the panoramas.
>
> Indeed, 2D MLLMs are rarely trained on panoramas. Therefore, we initially considered using skyboxes (i.e., 6-view images) instead of panoramas, and provided 12 images from both the current and previous scenes as input to the 2D MLLM. However, we observed that some 2D MLLMs struggle with processing multiple images. For example, DeepSeek generated incoherent responses when the number of input images exceeded six. We attribute this to the fact that these popular MLLMs are primarily pretrained to understand a small number of images. Therefore, we chose panoramas, as they can represent the holistic scene and preserve orientation despite distortion.
>
> To address your concern, we stitched the four surrounding views with the front view placed in the center, forming a single surround-view image as input to the one-shot InternVL model. As shown in the table below, there is only a negligible difference between using a panorama or a stitched multi-view image. We will include this analysis in the paper.
>
> | Input   | QA | Description | Rearrangement |
> |:-----------:|:------:|:---------------:|:-----------------:|
> | panorama     | 35.7   | 3.8             | 3.7               |
> | multi-view| 35.5   | 3.7             | 3.5               |

---

> > ### Comment · Reviewer_3iRY · 2025-08-04
> >
> > Thank you for the authors' rebuttal, which addresses most of the concerns. However, as Reviewer LpNV pointed out: “Also, it is suggested to add error analysis of previous methods and the proposed method to better reveal the points for improvement.” Since this is a benchmark paper, conducting a deeper analysis of why existing models (e.g., 2D MLLMs) fail in dynamic scenes is a crucial issue. If resources are limited, I recommend performing more detailed error analysis on a subset of the benchmark using large proprietary models (e.g., Gemini 2.5 or GPT-4) with chain-of-thought (CoT) prompting.

---

> > > ### Author Response · Authors · 2025-08-04
> > > **Response to Reviewer 3iRY**
> > >
> > > Dear Reviewer 3iRY,
> > >
> > > Thank you for your constructive feedback. We would like to address your concerns from two perspectives.
> > >
> > > **[Q1] Analysis of MLLM failures in dynamic scenario**
> > > >However, as Reviewer LpNV pointed out: “Also, it is suggested to add error analysis of previous methods and the proposed method to better reveal the points for improvement.” Since this is a benchmark paper, conducting a deeper analysis of why existing models (e.g., 2D MLLMs) fail in dynamic scenes is a crucial issue.
> > > The failure of few-shot 2D MLLMs and the absence of 3D MLLMs for multi-point clouds can be attributed to the lack of training data for dynamic scene understanding, and our work aims to bridge this gap.
> > >
> > > The failure of few-shot 2D MLLMs and the absence of 3D MLLMs for multi-point clouds can be attributed to the lack of training data for dynamic scene understanding, and our work aims to bridge this gap.
> > >
> > > Regarding why fine-tuned 2D MLLMs still fail in long-form tasks involving change description and rearrangement instructions, we can view the short QA tasks as decompositions of complex dynamic scene understanding tasks. The QAs are carefully distributed across egocentric and allocentric spatial information, as well as object properties.
> > > The underlying reasons can be identified by analyzing the performance results across different classes of fine-tuned models in Table 3 (analyzed in Lines 269-276). We observe that 2D MLLM performs better than 3D MLLMs in terms of egocentric information with the similar model size, while underperforming in allocentric reasoning and object property identification. This indicates that although panoramic views can effectively reconstruct visible areas and convey orientation information (Panocontext ECCV’14), they still suffer from occlusion issues and limited field of view constraints.
> > >
> > > **[Q2] Chain-of-Thought (CoT) prompting for deeper analysis**
> > > >If resources are limited, I recommend performing more detailed error analysis on a subset of the benchmark using large proprietary models (e.g., Gemini 2.5 or GPT-4) with chain-of-thought (CoT) prompting.
> > >
> > > Thank you for the suggestion. However, since GPT-4 is used to assist with dataset generation and evaluation, using it as a baseline could introduce bias (as noted in G-Eval, EMNLP'23). Additionally, the free version of Gemini has very low requests per day (RPD), and upgrading the subscription would take time that we cannot afford given our discussion deadline. Therefore, we provide results for long-form tasks using two 2D MLLMs with one-shot CoT prompting, compared to one-shot results without CoT, to analyze more deeply why they underperform in these tasks:
> > >
> > > | Model     | Description (wo CoT) | Description (with CoT) | Rearrangement (wo CoT) | Rearrangement (with CoT) |
> > > |-----------|----------------------|-------------------------|-------------------------|---------------------------|
> > > | InternVL  | 3.8                  | 5.3                     | 3.7                     | 4.9                       |
> > > | Janus     | 2.7                  | 3.6                     | 4.7                     | 4.3                       |
> > >
> > > We prompt the 2D MLLMs to separately analyze the two panoramic views in order to better capture allocentric information, then compare them based on the current situation. This strategy yields notable performance gains for both models, with the exception of Janus performing on-par for the rearrangement task, underscoring the importance of allocentric understanding. However, their performance still falls short compared to the parameter-efficient fine-tuned counterparts.
> > >
> > > The details of the CoT prompt will be included in the supplementary material. Fine-tuning with CoT introduces a promising direction for benchmarking, which we plan to explore in future work and will mention in the Future Work section.
> > >
> > > Please feel free to reach out if you have additional concerns.

---

> > > > ### Comment · Reviewer_3iRY · 2025-08-05
> > > >
> > > > Thanks for the authors’ rebuttal. I have no further questions.

---

> > > > > ### Author Response · Authors · 2025-08-05
> > > > > **Thank you**
> > > > >
> > > > > We appreciate your feedback and are glad to have addressed your concerns.

---

### Official Review · Reviewer_LpNV · 2025-07-24

**Rating:** 4
**Confidence:** 5

**Summary:**

The authors proposed the Situat3DChange dataset to evaluate situated change recognition ability. To construct Situat3DChange, 7 annotators annotated scene changes in detail, including plausible reasons for change, warnings of collision, change content, and guidance for rearrangement, resulting in 11K data points. After that, using an automatic method, the dataset was extended to three tasks including situated question answering, change description, along with rearrangement instruction for action generation from instruction, and covers 174K paired real-world scan pairs. The authors also proposed a method called SCReasoner, which uses Mamba for token selection and star operation for token fusion. The authors evaluated various VLMs on Situat3DChange in a zero-shot, one-shot manner, and fine-tuned two previous methods. The results show that fine-tuning on the Situat3DChange helps improve performance, and the proposed SCReasoner method obtained the highest accuracy on change description and rearrangement instruction tasks, and comparative accuracy on the QA task.

**Additional Feedback:**

The dataset is well structured, but I still have some concerns (see the weaknesses section). I am looking forward to the rebuttal response and am open to raising the score.

**Dataset Code Accessibility:**

Partly

**Dataset Code Comments:**

There is no code for dataset generation. It is suggested to have the generation code as well.

**Ethical Considerations:**

No, there are no or only very minor ethics concerns

**Final Justification:**

After reading the rebuttal and other reviewers' reviews, I decided to raise my score to Borderline accept.

**Limitations Weaknesses:**

- Some important studies in situated 3D reasoning are not mentioned in the paper. The authors should discuss how their work is different from the following papers:

[1] Song, C. H., Blukis, V., Tremblay, J., Tyree, S., Su, Y., & Birchfield, S. (2025). Robospatial: Teaching spatial understanding to 2d and 3d vision-language models for robotics.

[2] Yang, J., Yang, S., Gupta, A. W., Han, R., Fei-Fei, L., & Xie, S. (2025). Thinking in space: How multimodal large language models see, remember, and recall spaces.

[3] Ma, W., Ye, L., de Melo, C. M., Yuille, A., & Chen, J. (2025). Spatialllm: A compound 3d-informed design towards spatially-intelligent large multimodal models.

[4] Zhou, W., Tao, M., Zhao, C., Guo, H., Dong, H., Tang, M., & Wang, J. (2025). Physvlm: Enabling visual language models to understand robotic physical reachability.

- There is no ablation study regarding the structure of the proposed method, making it difficult to evaluate the structural choices of the proposed method. The reviewer suggests that the authors compare different modules for token selection and feature fusion.
- Some experimental details regarding previous methods are not well addressed in the paper, such as the input prompts for existing methods and the fine-tuning details of InternVL2 and LEO.
- The authors mainly used automatic evaluation metrics, such as BLEU and GPT; it is suggested to perform human evaluation on a dataset subset to analyze in depth how different the models are. Also, it is suggested to add error analysis of previous methods and the proposed method to better reveal the points for improvement.
- Although the authors have mentioned this in the limitations section, it would greatly enhance the paper to show transferability to other tasks, to confirm whether the models are learning domain-specific nuances or something more generally useful.

**Strengths Contributions:**

- The proposed dataset Situat3DChange deals with an important but less explored topic: understanding real-world dynamics. This kind of task is critical for various applications, such as robotic applications.
- An extensive evaluation of previous state-of-the-art VLMs is conducted, including zero-shot, one-shot, and fine-tuning setups.
- The experimental results seem to show that the performance of existing methods, along with the proposed method, is far from satisfactory, highlighting an important aspect for model improvement.
- The proposed method obtained good results on the proposed dataset, outperforming strong VLMs such as InternVL2.
- The proposed data is built upon human annotations, making it possible to evaluate model performance in human-like change understanding.

---

> ### Author Rebuttal · Authors · 2025-07-30
>
> Dear Reviewer LpNV,
>
> Thank you for recognizing the novelty and importance of the task we proposed in understanding dynamic situations and environments, as well as the human annotation and experimental efforts we invested in this work. Here are our responses to address your concerns.
>
> **[W1] Missing related work discussion**
> >Some important studies in situated 3D reasoning are not mentioned in the paper. The authors should discuss how their work is different from the following papers:
> [1] Song, C. H., Blukis, V., Tremblay, J., Tyree, S., Su, Y., & Birchfield, S. (2025). Robospatial: Teaching spatial understanding to 2d and 3d vision-language models for robotics.
> [2] Yang, J., Yang, S., Gupta, A. W., Han, R., Fei-Fei, L., & Xie, S. (2025). Thinking in space: How multimodal large language models see, remember, and recall spaces.
> [3] Ma, W., Ye, L., de Melo, C. M., Yuille, A., & Chen, J. (2025). Spatialllm: A compound 3d-informed design towards spatially-intelligent large multimodal models.
> [4] Zhou, W., Tao, M., Zhao, C., Guo, H., Dong, H., Tang, M., & Wang, J. (2025). Physvlm: Enabling visual language models to understand robotic physical reachability.
>
> Thank you for your suggestion and additional references. While these recent works are related to situated 3D reasoning, they differ significantly from our dataset, which addresses not only dynamic situations but also dynamic environments. We will include the recommended datasets in Table 1 and discuss them in the Related Work section.  "For robotic applications, RoboSpatial is a dataset that focuses on spatial context, object compatibility, and configuration. Phys100K is a large-scale multi-robot QA dataset. Thinking in Space introduces a dataset designed to facilitate spatial understanding based on short egocentric videos. SpatialLLM not only addresses spatial reasoning from a human perspective but also models object orientation relationships[...] In contrast to previous situational scene understanding datasets, we introduce Situat3DChange, a dataset for 3D scene understanding in dynamic scenarios and situations." We believe this would further highlight the contribution of our dataset in modeling the dynamic view in changing world.
>
> **[W2] Lack of ablation study**
> >There is no ablation study regarding the structure of the proposed method, making it difficult to evaluate the structural choices of the proposed method. The reviewer suggests that the authors compare different modules for token selection and feature fusion.
>
> We clarify that this work primarily focuses on dataset construction, contributing a new benchmark for situated 3D scene change understanding. Since no existing MLLM supports the task of comparing two similar point clouds, we propose the SCReasoner framework to avoid the token redundancy introduced by traditional token-prepend methods and reduce the input tokens of decoder. We explore simple projection strategies, including selective token projection using Mamba and simple channel-wise projection using linear layer, as well as parameter-free fusion methods such as element-wise addition and multiplication. As shown in the table below comparing projection and fusion modules, the combination of Mamba-based projection and element-wise multiplication achieves a significant gain over the LEO baseline (listed in the first row), demonstrating the effectiveness of our token-efficient framework. While we do not propose dedicated token selection or advanced feature fusion modules, further comparison could be out of scope, but this remains an interesting direction for future work, which we will briefly mention at the end.
>
> | Projection | Fusion | QA     | Description | Rearrangement |
> |:--------------:|:----------:|:----------:|:---------------:|:-----------------:|
> |-|-|51.9|12.7|30.1|
> | Linear         | Add.       |   52.8     |      12.6       |       30.3        |
> |                | Mul.       |   53.4     |      13.4       |       30.3        |
> | Mamba          | Add.       |   53.6     |      13.3       |       30.5        |
> |                | Mul.       | 53.8   |   13.9      |    30.7       |
>
> **[W3] Incomplete experimental details**
> >Some experimental details regarding previous methods are not well addressed in the paper, such as the input prompts for existing methods and the fine-tuning details of InternVL2 and LEO.
>
> The fine-tuning details are provided in Section C of the supplementary material.
> We will also provide the input prompts in the supplementary:
> ``` My current situation is: <situation> Here is a panoramic image from the previous scene: <image>. And here is a panoramic image from my current location: <image>. Based on this, please answer my question: <question> ```
>
> **[W4] Limited evaluation methodology**
> >The authors mainly used automatic evaluation metrics, such as BLEU and GPT; it is suggested to perform human evaluation on a dataset subset to analyze in depth how different the models are.
>
> We are grateful for your constructive feedback. We recruited four human evaluators who are not involved in this project and will acknowledge them in the paper. We selected SCReasoner, LEO, and one-shot InternVL for human evaluation. SCReasoner and LEO are compared to demonstrate our improvements, while one-shot InternVL represents an open-ended, training-free LLM. For each model, we randomly selected 40 samples for each QA type (excluding those related to distance) and each long-form task, resulting in the same 360 samples per model. As shown in the table, SCReasoner consistently outperforms LEO on the 360 sampled instances, with the performance gap further amplified when evaluated by human scores compared to GPT scores.
> | Model     | Description GPT | Description Human | Rearrangement GPT | Rearrangement Human | QA GPT | QA Human |
> |:-------------:|:-------------------:|:----------------------:|:----------------------:|:------------------------:|:----------:|:------------:|
> | InternVL      | 4.0                 | 6.5                    | 3.0                    | 7.5                      | 33.1       | 36.4         |
> | LEO           | 11.5                | 14.5                   | 22.5                   | 19.5                     | 43.3       | 45.6         |
> | SCReasoner| 14.0            | 20.5               | 26.0               | 31.5                 | 48.3   | 50.9     |
>
>
>
> Following OpenEQA (CVPR'24), we computed the Spearman correlation between human scores and GPT-generated scores. The GPT scores show a strong correlation with human evaluation (ρ > 0.6), indicating that GPT-based evaluation aligns well with human judgment.
>
> | | Description | Rearrangement | QA  | Average |
> |:----------------:|:---------------:|:-----------------:|:-------:|:-------:|
> | Spearman Corr. |      0.75       |       0.70        |  0.94   |  0.80   |
>
>
> **[W5] Lack of error analysis**
> >Also, it is suggested to add error analysis of previous methods and the proposed method to better reveal the points for improvement.
>
> We appreciate the insightful feedback and will include the run-to-run error analysis in the camera-ready version. In our original submission, we followed LEO’s setting by fixing the random seed. To better analyze the errors and demonstrate the performance gain of SCReasoner over LEO, we conducted three additional runs with different random seeds, resulting in four runs in total. The mean scores and standard deviations for each task are reported below, showing that SCReasoner consistently outperforms LEO.
> |              | QA             | Description     | Rearrangement    |
> |--------------|--------------------|----------------------|-----------------------|
> | LEO      | 52.760 ± 0.511     | 12.733 ± 0.117       | 30.165 ± 0.410        |
> | SCReasoner | 53.420 ± 0.232     | 13.629 ± 0.287       | 30.753 ± 0.216        |
>
> **[W6] Unaddressed transferability**
> >Although the authors have mentioned this in the limitations section, it would greatly enhance the paper to show transferability to other tasks, to confirm whether the models are learning domain-specific nuances or something more generally useful.
>
> Thank you for your suggestions. However, this appears to be a misunderstanding, and we will revise the Limitations section to clarify the issue. Transferability is already discussed in Lines 290–303. As shown in Tables 4 and 5, our dataset generalizes well to tasks in a different domain (i.e., ScanNet-based datasets). Training with Situt3DChange, alongside ScanNet data, enhances the model’s understanding of both static and dynamic scenes.
>
> The actual limitation lies in the data generation pipeline, which is not transferable due to its reliance on human annotations. Moreover, since ground truth annotations for the 3RScan test set are unavailable, our dataset is primarily suitable for training multimodal large language models with hold-out evaluation, rather than task-specific small models.
>
> **[W7] Lack of dataset generation code**
> >There is no code for dataset generation. It is suggested to have the generation code as well.
>
> The prompt used for data generation is detailed in Section A of the supplementary material. Additionally, examples of the raw input data will be provided in our GitHub repository, allowing followers to replicate and experiment with the dataset generation process.

---

> > ### Author Response · Authors · 2025-08-04
> > **To Reviewer LpNV**
> >
> > Dear Reviewer LpNV,
> >
> > Thank you for your effort during the review procedure. Since the author-reviewer discussion will end soon, we would like to kindly ask if you have any questions about the rebuttal. We would like to know what we can do to facilitate further discussions between us. Thank you for your attention to this matter. We greatly value your guidance and are happy to assist in any way to facilitate the process.

---

> > ### Comment · Reviewer_LpNV · 2025-08-05
> > **The rebuttal addressed most of my concerns**
> >
> > The rebuttal addressed most of my concerns. But I still believe having a deeper discussion on the weaknesses of existing multimodal LLMs would be valuable, but as suggested by authors, it would be difficult to incorporate them during the discussion period. I have no further questions. Thanks again for the rebuttal.

---

> > > ### Author Response · Authors · 2025-08-05
> > > **Further clarification on the deeper discussion**
> > >
> > > Dear Reviewer LpNV,
> > >
> > > Thank you for your time in reviewing our work. We would like to clarify that, in accordance with your review, we have added error analysis of previous methods and our proposed method to better reveal the points for improvement. We conducted several run-to-run experiments with our SCReasoner and our 3D MLLM baseline, LEO, to validate the improvement of our method.
> > > |              | QA             | Description     | Rearrangement    |
> > > |--------------|--------------------|----------------------|-----------------------|
> > > | LEO      | 52.760 ± 0.511     | 12.733 ± 0.117       | 30.165 ± 0.410        |
> > > | SCReasoner | 53.420 ± 0.232     | 13.629 ± 0.287       | 30.753 ± 0.216        |
> > >
> > > Regarding our response to reviewer 3iRY, we delved into discussing the reasons why 2D MLLMs fail in this task: concise QA can be regarded as a decomposition of complex dynamic scene understanding. While 2D MLLMs perform superior in understanding egocentric information, they still fall short in understanding allocentric information and object properties. CoT prompting analysis has been conducted, showing that prompting for allocentric scene understanding improves the performance of one-shot MLLMs.
> > >
> > > | Model     | Description (wo CoT) | Description (with CoT) | Rearrangement (wo CoT) | Rearrangement (with CoT) |
> > > |-----------|----------------------|-------------------------|-------------------------|---------------------------|
> > > | InternVL  | 3.8                  | 5.3                     | 3.7                     | 4.9                       |
> > > | Janus     | 2.7                  | 3.6                     | 4.7                     | 4.3                       |
> > >
> > > Time constraints prevented us from upgrading our Gemini subscription, but we are still open to further discussion. We would like to include the one-shot and CoT results for Gemini in the camera-ready version.
> > >
> > > If you believe we still lack any other discussion points, please feel free to suggest them. Thanks again.

---

### Note · Authors · 2025-08-12

Dear AC, Reviewers,

Thank you for your efforts in improving our work. Having addressed all reviewer concerns, we summarize the rebuttal and discussion phases as follows.

**Strengths Recognized by All Reviewers**

The initial review yielded an average score indicating acceptance. We are pleased that all reviewers recognized the originality and novelty of our work:

- Situat3DChange tackles an **important yet underexplored challenge of understanding real-world dynamics** through both human situational awareness and environmental changes.

- The **three tasks (QA, Description, and Rearrangement) are well-structured and large-scale**, with 121K, 36K, and 17K samples respectively, facilitating human-AI collaboration **across the perception-to-action spectrum**.

- The **extensive expert annotations** (11K) ground the dataset in genuine human perception.

- The **proposed SCReasoner** is more **effective** than baselines **while being token-efficient**.

- **Extensive experiments** on the benchmark reveal the performance of **existing MLLMs under different setups**.

**Post-Review Revisions**

- **Additional material will be provided to solidify the benchmark**, including human evaluation of MLLM predictions to establish GPT-score reliability, statistics on QA categories by change relevance, multiple-run results confirming SCReasoner's consistent improvement over LEO, and Chain-of-Thought experiments to explore existing MLLM weaknesses.

- We will **enhance literature discussion** to improve research quality by adding more citations to verify our work's novelty against existing situated 3D reasoning works, and incorporating cognitive science references to explain our use of inconsistent direction and distance elements for perception and action.

- Several points require **further clarification**: token efficiency claims throughout the paper, differences between SCReasoner and LEO with architectural variations in Section 4, rationale for using panoramic views rather than multi-view images in Section 4, analysis of concise QA results concerning complex dynamic reasoning decomposition in Section 5, transferability in Section 6, and details on inference prompts, panorama generation, data generation, and experimental details in supplementary materials.

**Overall, all reviewers acknowledged the dataset's valuable contribution, and we can confidently address all revisions in the camera-ready version.**

Best regards,

Authors

---

### Decision · Program_Chairs · 2025-09-18

**Decision:**

Accept (poster)

**Comment:**

The paper proposes Situat3DChange, a dataset and benchmark for studying changes in a 3D environment.  The dataset is built on 3DRScan and consists of 904 scenes paired with questions about what has changed and what actions is needed to revert the changes to form a dataset with 174K data instances.  The dataset is constructed using a combination of human annotation and text-generation using GPT-4.  The main contributions of the work include 1) the Situate3DChange dataset, 2) a model to tackle the task (SCReasoner), 3) experiments comparing VLMs with the proposed model.

The paper was reviewed by four experts in the area, and all four reviewers are positive on the work noting that 1) the task of studying changes in 3D scenes is interesting and understudied, 2) the dataset is large-scale and rich, 3) the methology for creating the dataset is well explained, 4) the proposed SCReasoner is an intuitive approach, and 5) the experiments are comprehensive.

Despite the strengths of the work, reviewers did note some areas of improvement including:
- Missing related work [fa1y, LpNV]
- Missing ablation study [LpNV]
- Missing experimental details [LpNV]
- Limited details on some design choices [fa1y]
- Limited details about the some aspects of the composition of the questions in the dataset [fa1y]
- Limited details about the SCReasoner [3iRY]
The authors did a great job during the rebuttal period to provide clarifications and additional results.

The AC agrees with the positive ratings from the reviewers, and recommends acceptance.  The authors should incorporate clarifications and responses to reviewer questions in the camera ready.

===== FINAL UPDATE FROM DB Track PCs ====

The final decision for this paper has been taken by the program chairs after consultation with the SACs. All Senior Area Chairs have ranked papers according to the feedback from the AC during the review process. We decided to leave the original meta-review to reflect the opinion of the AC in light of the initial discussions with reviewers and SAC.